# Asymmetric localization of the cell division machinery during *Bacillus subtilis* sporulation

Kanika Khanna[†], Javier Lopez-Garrido[‡], Joseph Sugie, Kit Pogliano*, Elizabeth Villa*

Division of Biological Sciences, University of California San Diego, La Jolla, United States

**Abstract** The Gram-positive bacterium *Bacillus subtilis* can divide via two modes. During vegetative growth, the division septum is formed at the midcell to produce two equal daughter cells. However, during sporulation, the division septum is formed closer to one pole to yield a smaller forespore and a larger mother cell. Using cryo-electron tomography, genetics and fluorescence microscopy, we found that the organization of the division machinery is different in the two septa. While FtsAZ filaments, the major orchestrators of bacterial cell division, are present uniformly around the leading edge of the invaginating vegetative septa, they are only present on the mother cell side of the invaginating sporulation septa. We provide evidence suggesting that the different distribution and number of FtsAZ filaments impact septal thickness, causing vegetative septa to be thicker than sporulation septa already during constriction. Finally, we show that a sporulation-specific protein, SpoIIE, regulates asymmetric divisome localization and septal thickness during sporulation.

**\*For correspondence:**
kpogliano@ucsd.edu (KP);
evilla@ucsd.edu (EV)

**Present address:** [†]Department of Bioengineering and ChemH, Stanford University, Stanford, United States; [‡]Max Planck Institute for Evolutionary Biology, Plön, Germany

**Competing interests:** The authors declare that no competing interests exist.

## Introduction

Bacterial cell division involves the invagination of the cellular membrane(s) and peptidoglycan (PG) cell wall to split the cell into two progeny cells. In most bacteria, cell division is initiated when the tubulin homologue and GTPase, FtsZ, polymerizes to form a ring-like structure, called the Z-ring, at the division site (*Bi and Lutkenhaus, 1991*; *de Boer et al., 1992*; *Mukherjee and Lutkenhaus, 1994*). The Z-ring serves as a scaffold onto which about a dozen other proteins assemble to form a mature cell division machinery called 'divisome' (*Haeusser and Margolin, 2016*). In the Gram-positive bacterium *Bacillus subtilis*, divisome assembly proceeds in two steps. First, proteins that interact directly with FtsZ and promote its association with the membrane are recruited to the divisome (*Gamba et al., 2009*). One essential protein recruited at this stage is FtsA, an actin homologue which tethers FtsZ to the membrane via its conserved C-terminal amphipathic helix (*Pichoff and Lutkenhaus, 2005*; *Szwedziak et al., 2014*). FtsA can also form protofilaments via polymerization in an ATP-dependent manner (*Szwedziak et al., 2012*). Second, proteins involved in septal PG synthesis and remodeling are recruited to the divisome (*Gamba et al., 2009*). Recent studies suggest that treadmilling by FtsZ filaments serves as a platform to drive the circumferential motion of septal PG synthases in both *B. subtilis* and *Escherichia coli*, suggesting that this dynamic property of FtsZ filaments can regulate cell wall remodeling in the septal disc (*Bisson-Filho et al., 2017*; *Yang et al., 2017*).

*B. subtilis* is a unique model system to study cell division as it can divide via two modes (*Figure 1*). During vegetative growth, the septum is formed at the midcell, producing two daughter cells of equal sizes (*Wang and Lutkenhaus, 1993*). However, during sporulation, the division septum is formed closer to one pole to produce a smaller forespore and a larger mother cell. Subsequently,

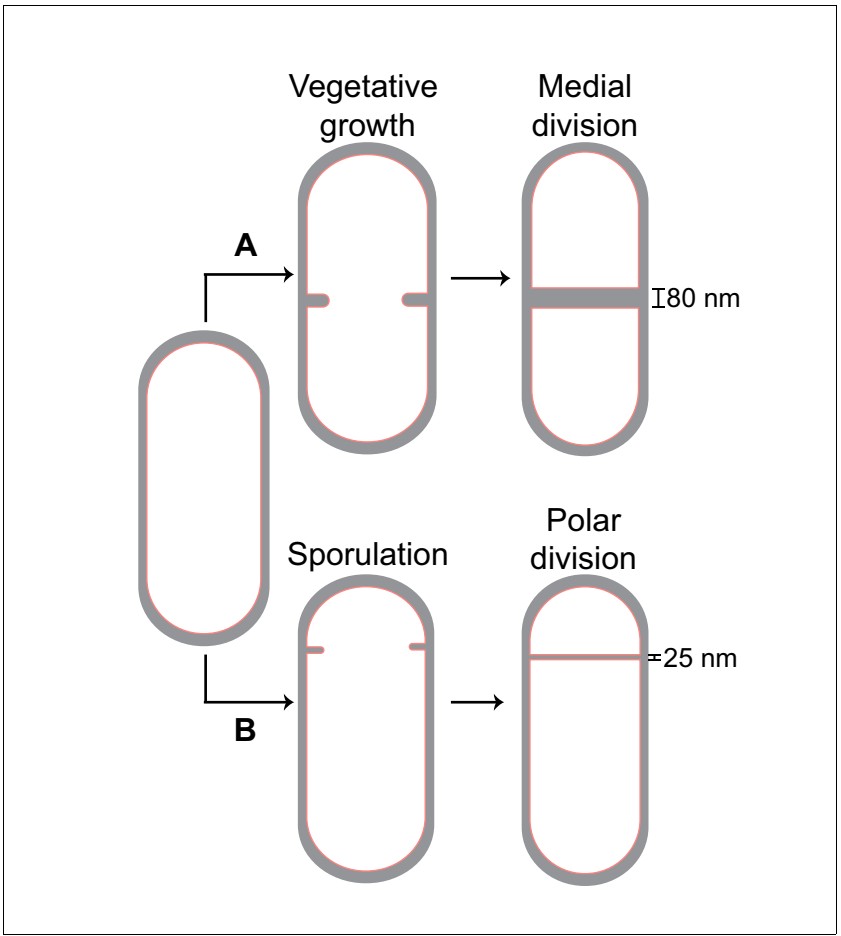

**Figure 1.** Cell division in *Bacillus subtilis.* Schematic of cell division in *B. subtilis* during (**A**) vegetative growth and (**B**) sporulation. The thickness of septa upon their closure is indicated for both cases.

the forespore is engulfed by the mother cell, producing an internal endospore, that upon maturation, is released by mother cell lysis (*Khanna et al., 2020*; *Riley et al., 2021b*; *Tan and Ramamurthi, 2014*). Vegetative and sporulating cells differ not only in the positioning of the division site but also in the thickness of their septa, with the medial vegetative septum being almost four times thicker than the polar sporulating septum (~80 nm vs. ~25 nm) (*Illing and Errington, 1991*; *Tocheva et al., 2013*). Another distinction is the presence of a membrane protein SpoIIE, which colocalizes with and directly interacts with FtsZ in sporulating cells (*Arigoni et al., 1995*; *Levin et al., 1997*; *Lucet et al., 2000*), but is absent in vegetative cells. Recent evidence suggests that SpoIIE localizes only on the forespore side of the sporulation septum at the onset of membrane constriction before being released into the forespore membrane upon septum formation (*Carniol et al., 2005*; *Errington, 2003*; *Eswaramoorthy et al., 2014*; *Guberman et al., 2008*; *Wu et al., 1998*). Certain *spoIIE* mutants also form thicker polar septa compared to wild-type sporangia (*Barák and Youngman, 1996*; *Illing and Errington, 1991*). These findings posit that there may be differences in the divisome organization between the vegetative and the sporulation septum that play a role in regulating septal thickness. However, a mechanistic understanding of how cell division is regulated during *B. subtilis* sporulation remains elusive.

In this study, we used cryo-focused ion beam milling coupled with cryo-electron tomography (cryo-FIB-ET) to reveal the molecular architecture of the cell division machinery in *B. subtilis* during vegetative growth and sporulation. Our results demonstrate that FtsAZ filaments have distinct spatial organization during the two modes of division and that SpoIIE regulates the positioning of FtsAZ filaments during sporulation. We further provide evidence that this distinct organization leads to a thinner polar septum during sporulation. Our results pave the way for future studies in the field of

bacterial cell division, giving rise to testable hypothesis on how bacteria regulate protein localization in space and time to carry out the critical process of cytokinesis under different gene expression programs.

## Results

### FtsAZ filaments in vegetative *B. subtilis*

Cryo-electron tomography (cryo-ET) is a method to obtain three-dimensional reconstructions of biological specimens. Previously, cryo-ET images of *E. coli* and *Caulobacter crescentus* revealed the presence of a series of electron-dense puncta (dots) at the division site corresponding to cross-sectional views of FtsZ filaments encircling the cell (*Li et al., 2007*; *Szwedziak et al., 2014*). However, because of their increased thickness, *B. subtilis* cells were precluded from high-resolution tomography until recently, when we incorporated a cryo-FIB milling step in the workflow to provide mechanistic details about engulfment in *B. subtilis* sporangia (*Khanna et al., 2019*; *Lopez-Garrido et al., 2018*; *Figure 2*, *Figure 2—figure supplement 1*, see also Materials and methods).

To obtain insights into the divisome architecture in *B. subtilis*, we first observed the leading edge of the invaginating septum in cryo-FIB-ET images of dividing vegetative cells (*Figure 3*, *Figure 3— figure supplement 1*). Our data revealed two series of dots at the division site that were distributed uniformly along the boundary of the leading edge: a membrane-proximal series of dots (~6.5 nm from the invaginating membrane) and a membrane-distal series of dots (~14 nm from the invaginating membrane) (*Figure 3A, B, D, F*, *Figure 3—figure supplement 1A, B*, *Figure 3—video 1*).

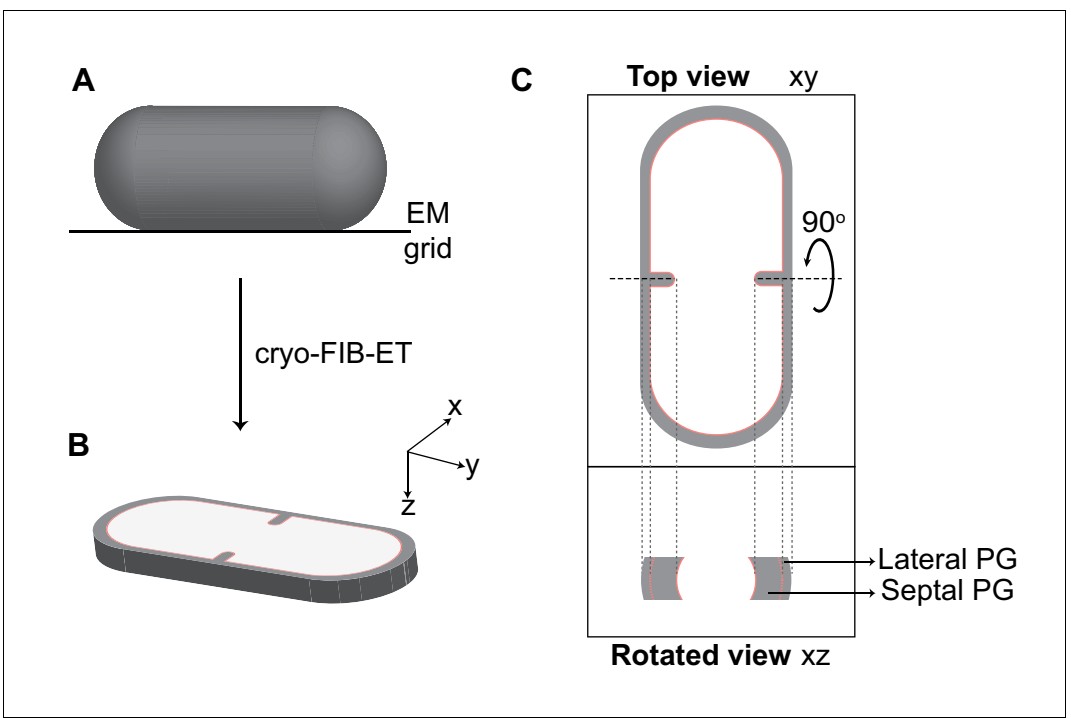

**Figure 2.** Visualization of cellular cross-sections in dividing cells. Schematic explaining visualization of cells in different planes in three-dimension (3D). (**A**) Initially, the rod-shaped *Bacillus* lies flat on an electron microscopy (EM) grid. (**B**) Representation of 3D view of a cellular section obtained by cryo-FIB-ET in the xyz coordinate axis. x axis represents the length along the short axis of the cell, y axis represents the length along the long axis of the cell and z axis represents the height of the cellular specimen. (**C**) Top panel: projection image of the cell in the xy coordinate plane (top view). Bottom panel: the corresponding projection image in the xz coordinate plane when the cell is rotated about its short axis by 90˚ (side/rotated view). The lateral and septal peptidoglycan (PG) are also indicated. See also *Figure 2—figure supplement 1*.

The online version of this article includes the following figure supplement(s) for figure 2:

**Figure supplement 1.** Cryo-FIB milling workflow and analysis.

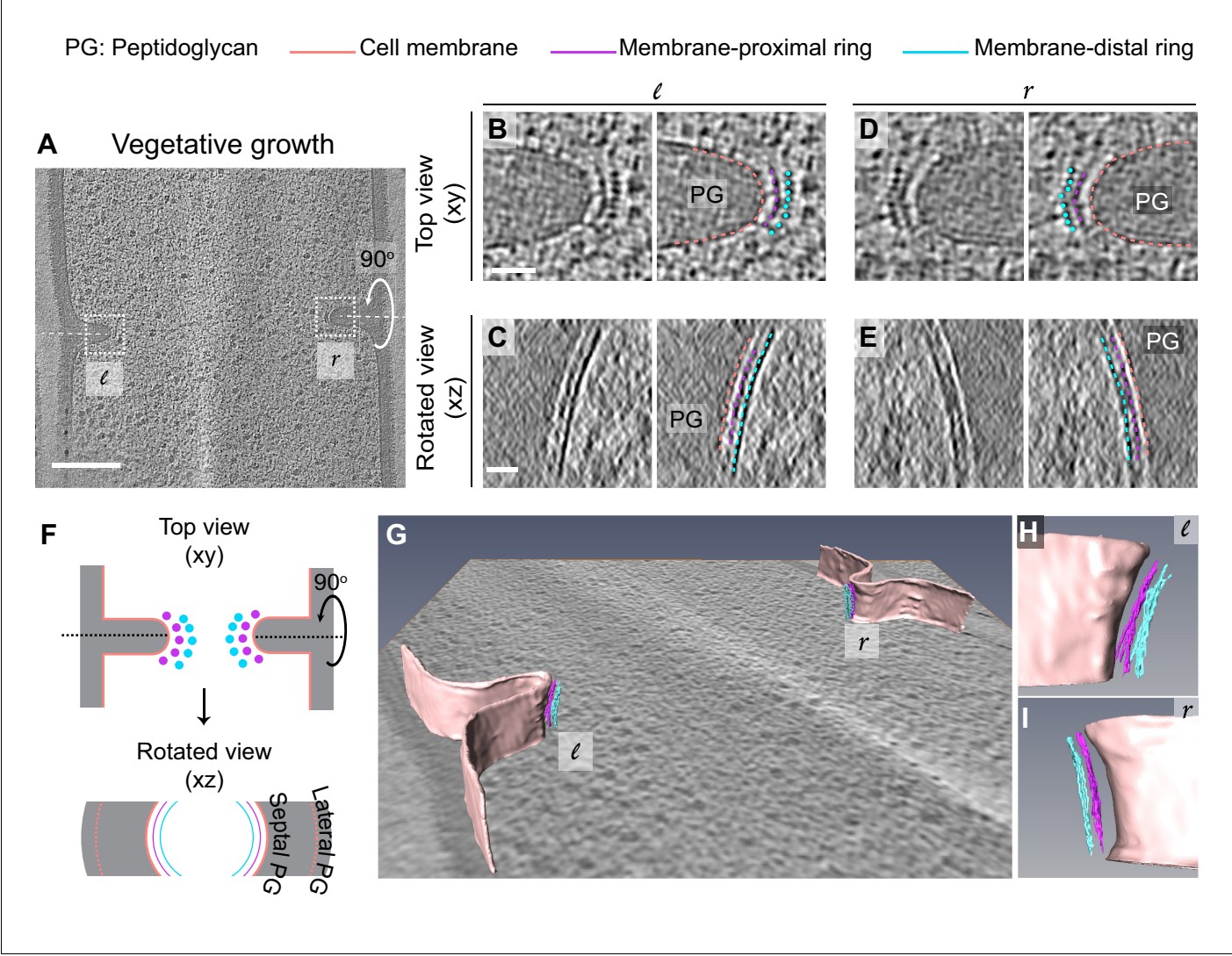

**Figure 3.** Architecture of divisome in vegetative *B. subtilis* cells. (**A**) Slice through a tomogram of a dividing vegetative cell. The insets (*l* for left and *r* for right side of the septum) highlight the leading edges of the invaginating septum. (**B**) Left panel: zoomed-in view of the '*l*' inset in (**A**) in the xy coordinate plane. Right panel: same as left with peptidoglycan (PG; gray), cell membrane (peach), membrane-proximal series of dots (pink) and membrane-distal series of dots (blue) highlighted. The same color scheme is followed throughout. (**C**) Left panel: view of the septal disc corresponding to (**B**) in the xz coordinate plane obtained by rotating the cell around its short axis by 90°. Right panel: same as left with PG, cell membrane, membrane-proximal ring and membrane-distal ring highlighted. (**D**) Left panel: zoomed-in view of the inset '*r*' in (**A**) in the xy coordinate plane. Right panel: same as left with cellular parts highlighted. (**E**) Left panel: view of the septal disc corresponding to (**D**) in the xz coordinate plane obtained by rotating the cell around its short axis by 90°. Right panel: same as left with cellular parts highlighted. (**F**) Schematic of the arrangement of the cytoskeletal machinery in dividing vegetative cells as seen in the xy and xz coordinate planes. All cellular parts detailed previously are highlighted in the same color scheme. (**G**) Annotation of the cell membrane and filaments corresponding to the membrane-proximal and the membrane distal rings for the tomogram shown in (**A**). (**H** and **I**) represent zoomed-in views of the left (I) side and the right (r) side of the invaginating septum of the segmentation in (**G**), respectively. Scale bars: (**A**) 200 nm, (**B–E**) 25 nm. Scale bars are omitted from (**G–I**) owing to their perspective nature. See also *Figure 3—figure supplements 1* and *2*. * A note regarding annotation of cytoskeletal filaments. In this and in subsequent figures, we have annotated membrane-distal bundle with dots in xy views, and in slices where we could not resolve them unambiguously, we have depicted them as dashed lines. We have depicted membrane-proximal bundle as dashed lines in xy views since in many instances it was not possible to resolve individual dots corresponding to the membrane-proximal bundle with full certainty. We used dashed lines to represent both the membrane-distal and the membrane-proximal cytoskeletal rings in rotated views (xz/yz slices) due to missing wedge issues and the quality of tomograms at times that made it difficult to assign continuity at the pixel level for all images.

The online version of this article includes the following video and figure supplement(s) for figure 3:

**Figure supplement 1.** Cytoskeletal filaments in vegetative cells during exponential growth.

**Figure supplement 2.** Intensity of cytoskeletal filaments in vegetative cells.

*Figure 3 continued on next page*

*Figure 3 continued*
**Figure 3—video 1.** Series of slices through the cryo-electron tomogram of a dividing vegetative *B. subtilis* cell shown in *Figure 3A*.
https://elifesciences.org/articles/62204#fig3video1
**Figure 3—video 2.** Series of slices through the cryo-electron tomogram of a dividing vegetative *B. subtilis* cell shown in *Figure 3A* when the cell is rotated about its short axis by 90°.
https://elifesciences.org/articles/62204#fig3video2
**Figure 3—video 3.** Series of slices through the cryo-electron tomogram of a dividing vegetative *B. subtilis* cell shown in *Figure 3A* wherein the cell membrane (peach), FtsA ring (pink) and FtsZ ring (blue) are annotated as in *Figure 3G–I*.
https://elifesciences.org/articles/62204#fig3video3

Rotation of the 3D volume of the tomogram around the short axis of the cell (*Figure 2*) demonstrated that each series of dots corresponded to filamentous structures of ~3.5 nm diameter arranged next to each other, forming a bundle that spans the circumference of the cell (*Figure 3C, E–I*, *Figure 3—figure supplement 1C*, *Figure 3—video 2*). In the membrane-distal series of dots, we could resolve individual filaments spaced ~5.5 nm apart while the filaments corresponding to the membrane-proximal series of dots were more diffuse (*Figure 3B, D*). In a few data sets, we observed densities that connect the membrane-proximal and the membrane-distal bundles in a ladder-like arrangement (*Figure 4*, indicated by white lines in *Figure 4B, D*). We do not exactly know which factors may produce the ladder-like arrangements but other FtsZ-binding proteins including SepF, ZapA or EzrA are potential candidates to test in the future (*Duman et al., 2013*; *Gueiros-Filho and Losick, 2002*; *Singh et al., 2007*). In cryo-ET, contrast in images reflects variation in mass density across the biological specimen. Analysis of the intensities of the two rings revealed that the membrane-distal ring is denser and more continuous than the membrane-proximal ring (*Figure 3—figure supplement 2*).

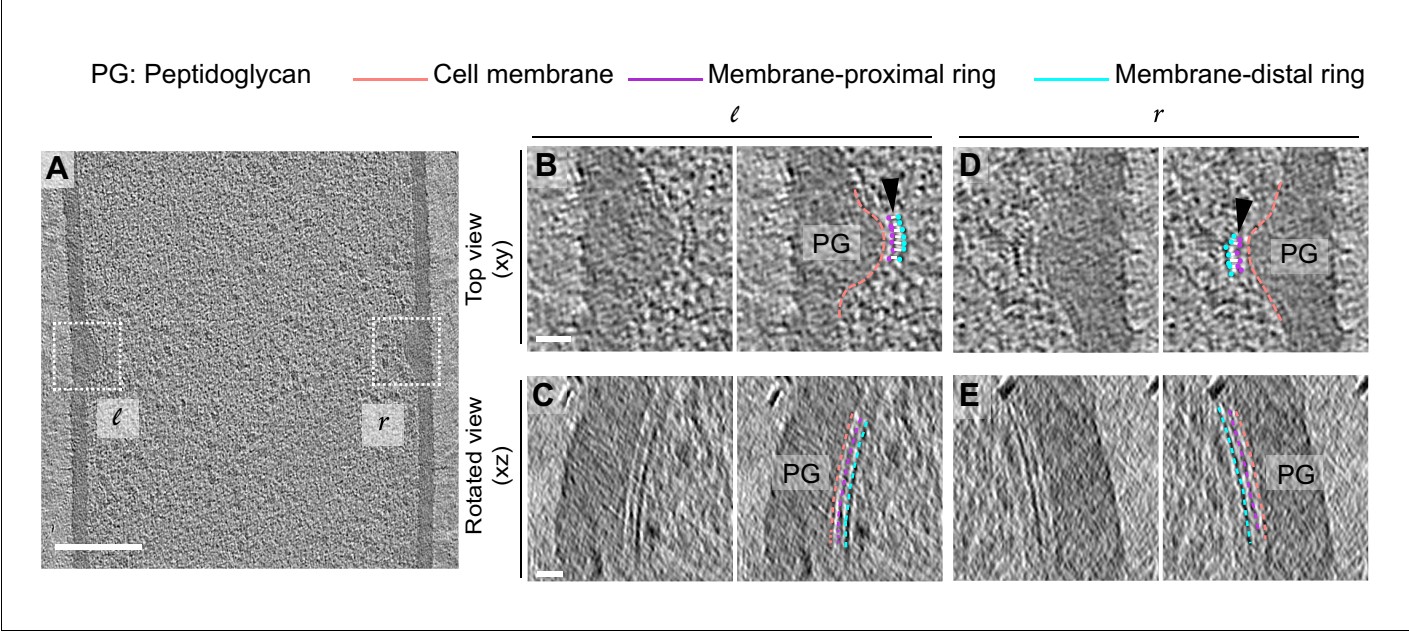

**Figure 4.** Architecture of bridging densities in the divisome of vegetative *B. subtilis* cells. (A) Slice through a tomogram of a dividing vegetative cell. The insets (*l* for left and *r* for right side of the septum) highlight the leading edges of the invaginating septum. (B) Left panel: zoomed-in view of the '*l*' inset in (A) in the xy coordinate plane. Right panel: same as left with peptidoglycan (PG; gray), cell membrane (peach), membrane-proximal series of dots (pink) and membrane-distal series of dots (blue) highlighted. 'Ladder-like' connection between the membrane-proximal and the membrane-distal bands is shown by white lines (indicated by a black arrow). The same color scheme is followed throughout. (C) Left panel: view of the septal disc corresponding to (B) in the xz coordinate plane obtained by rotating the cell around its short axis by 90°. Right panel: same as left with PG, cell membrane, membrane-proximal ring and membrane-distal ring highlighted. (D) Left panel: zoomed-in view of the inset '*r*' in (A) in the xy coordinate plane. Right panel: same as left with cellular parts highlighted. (E) Left panel: view of the septal disc corresponding to (D) in the xz coordinate plane obtained by rotating the cell around its short axis by 90°. Right panel: same as left with cellular parts highlighted. Scale bars: (A) 200 nm, (B–E) 25 nm.

## FtsA and FtsZ comprise the membrane-proximal and the membrane-distal ring, respectively

Our data suggest that the arrangement of the two cytoskeletal rings that mediate cytokinesis in *B. subtilis* differs from that of Gram-negative bacteria like *E. coli* and *C. crescentus* where only a single bundle of FtsZ filaments is visible at a distance of ~16 nm from the inner membrane (*Li et al., 2007*; *Szwedziak et al., 2014*). In *E. coli*, another ring composed of FtsA filaments is visible at a distance of ~8 nm from the membrane only when FtsA is overexpressed, such that the ratio of FtsZ to FtsA alters from 5:1 in wild type to 1:1 in the modified strain (*Szwedziak et al., 2014*). Based on these data and the knowledge that both FtsA and FtsZ form filaments in vivo and in vitro (*Mukherjee and Lutkenhaus, 1994*; *Szwedziak et al., 2012*), we hypothesized that in *B. subtilis* the membrane-proximal ring is composed of FtsA filaments and the membrane-distal ring that of FtsZ filaments.

To unambiguously establish the identity of the two rings, we constructed a *B. subtilis* strain with an extra glutamine-rich linker region between the globular N-terminal domain of FtsZ and its C-terminal helix that binds FtsA (hereafter referred to as FtsZ-linker$_{Q-rich}$; *Figures 5* and *6*, *Figure 5—figure supplement 1*). In a similar *E. coli* mutant strain, the Z-ring was further from the membrane by ~5 nm compared to the wild type (*Szwedziak et al., 2014*). In *B. subtilis* FtsZ-linker$_{Q-rich}$, the distance of the membrane-proximal ring from the invaginating membrane remained unaltered compared to wild type, while the membrane-distal filaments were further from the membrane compared to wild type (17.5 ± 1.1 nm vs. 14 ± 0.6 nm) with a wide distribution ranging from ~15 to ~19 nm (*Figure 5B*, *Figure 5—source data 1*), establishing the membrane-distal filaments to be FtsZ.

In a few instances, we detected 'stray' filaments at a distance of ~12–20 nm from the membrane and away from the leading edge (black arrows, *Figure 6A, B, D–G*). We speculate that such filaments correspond to FtsZ but without a corresponding FtsA filament to tether it to the membrane as the incorporation of the long linker in the C-terminal of FtsZ likely affects the interaction between the two proteins, as also evidenced by longer filamented cells in the presence of the linker (*Figure 5—figure supplement 1C, D*). This is substantiated by previous studies showing that *B. subtilis* can grow without FtsA, albeit slowly and in a filamentous manner (*Beall and Lutkenhaus, 1992*). Instead, in *B. subtilis*, FtsZ may use an alternate anchor like SepF or EzrA to bind to the membrane in the absence of FtsA as these divisome proteins also have an N-terminal domain that binds to the membrane and a C-terminal domain that interacts with FtsZ (*Duman et al., 2013*; *Singh et al., 2007*). In a few tomograms, we also observed doublets/triplets of individual filaments only in the membrane-distal band, an arrangement that was previously reported for FtsZ filaments in tomograms of dividing *E. coli* cells (*Szwedziak et al., 2014*) (blue/orange arrows, *Figure 6H–L*), further demonstrating that only the membrane-distal ring corresponded to the Z-ring. In one tomogram, we also observed an additional faint ring between the membrane-proximal and the membrane-distal rings (~14 nm from the membrane) that might correspond to few FtsZ filaments that localize closer to the membrane likely due to flexibility of the linker domain (in green, *Figure 6—figure supplement 1*). We assigned the identity of the membrane-proximal ring to be FtsA filaments as no other cell division proteins form filaments in vivo or in vitro and its distance to the membrane is comparable to that of FtsA filaments in *E. coli* upon overexpression (*Szwedziak et al., 2014*). To further establish the identity of the filaments, we verified that the filament thickness, continuity along the septal plane and distribution of intensity values of the rings in FtsZ- linker$_{Q-rich}$ compared to the wild type were all similar (*Figure 6—figure supplement 2*). Thus, we conclude that the more continuous membrane-distal ring of FtsZ filaments is tethered to the membrane via a patchy membrane-proximal ring of FtsA filaments (*Figure 5C*, *Figure 3—video 3*).

Our cryo-FIB-ET data indicated that the Z-ring was mostly continuous in the cellular sections we imaged. We observed fairly continuous Z-rings in 9 out of 16 tomograms of dividing vegetative cells that we captured (*Figures 3C, E* and *4C, E*, *Figure 3—figure supplement 1C*, *Figure 3—video 2*). We could not reliably assess the continuity of the Z-ring in the remaining seven tomograms because they were either too thick or did not yield high-resolution data due to their orientation with respect to the tilt axis. It is also difficult to ascertain with confidence whether the Z-ring is continuous throughout the cellular volume since we are only sampling a section of the cell (~200 nm of ~1.2 μm) due to ablation by cryo-FIB milling and due to missing wedge issue associated with cryo-ET data collection (*Figure 2—figure supplement 1D*). However, the data from our highest quality tomograms,

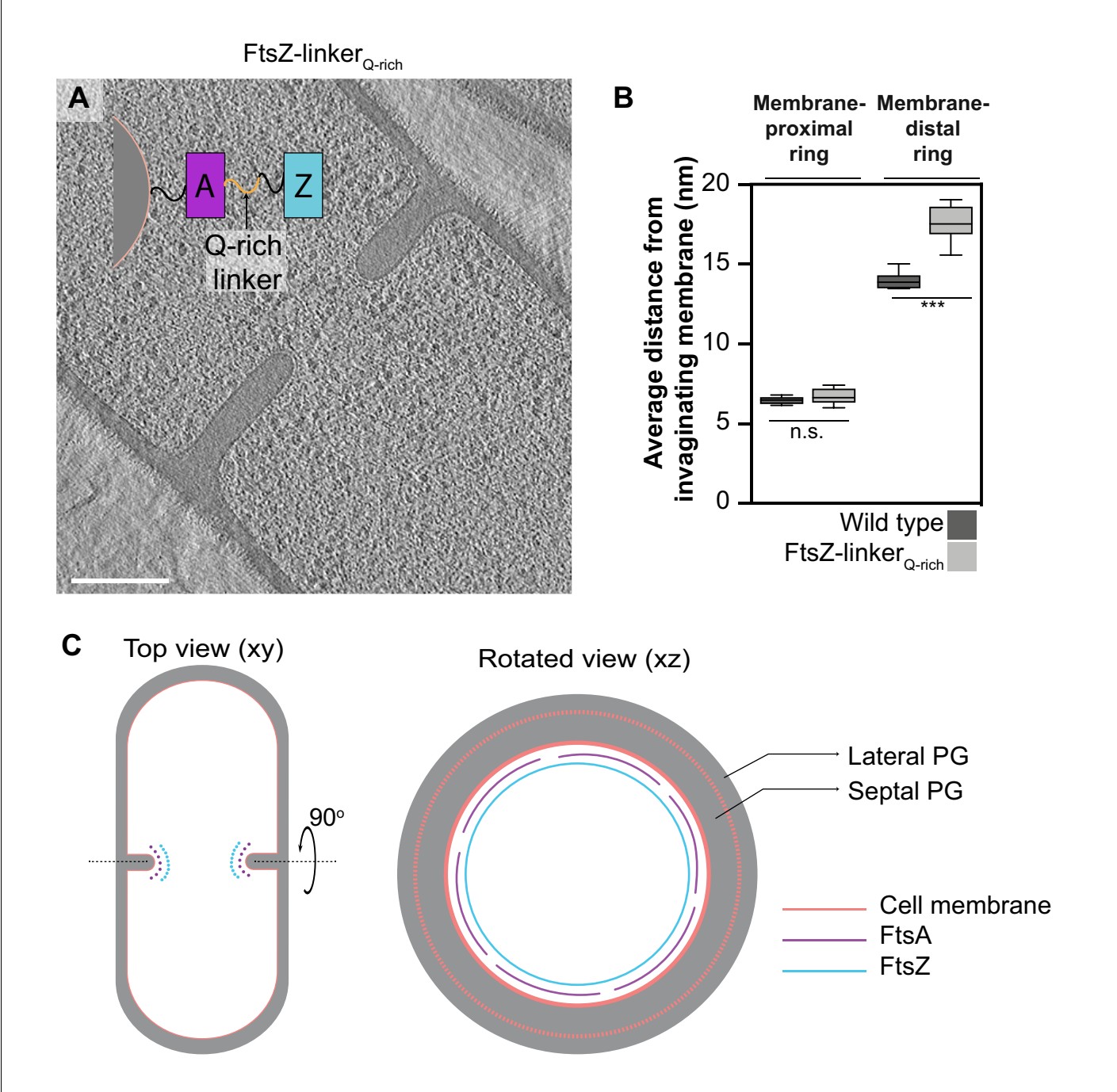

**Figure 5.** Identity of cytoskeletal filaments. (**A**) Slice through a tomogram of FtsZ-linker_Qrich dividing vegetative cell. A schematic showing the construction of the modified strain is overlaid on the tomogram slice wherein peptidoglycan (PG; gray) and cell membrane (peach) are highlighted. FtsZ (blue) is tethered to the membrane via FtsA (pink), and the two interact via a linker region (black + Q-rich linker in orange). Same color scheme is followed throughout. (**B**) Box-plot showing the distance of the membrane-proximal and the membrane-distal rings from the cell membrane in wild type and FtsZ-linker_Qrich strains. Error bars indicate standard deviation (n.s.: p>0.05, ***p≤0.001, unpaired t-test). (**C**) Schematic illustrating the arrangement and identity of the cytoskeletal machinery with PG (gray), cell membrane (peach), FtsZ (blue) and FtsA (pink) highlighted. In the top view (xy coordinate plane), the cytoskeletal machinery is visible as two series of dots at the nascent septum. In the rotated view (xz coordinate plane), denser and more continuous ring formed by FtsZ is tethered to the membrane via a patchy ring formed by FtsA filaments. See also *Figure 5—figure supplement 1*, *Figure 5—source data 1*.

The online version of this article includes the following source data and figure supplement(s) for figure 5:

**Source data 1.** Distance of the cytoskeletal rings (membrane-proximal FtsA and membrane-distal FtsZ) from the invaginating cellular membrane.

*Figure 5 continued on next page*

*Figure 5 continued*

**Figure supplement 1.** FtsZ-linker$_{Q-rich}$ phenotype.

each of which capture different parts of the septal disc, consistently show that the Z-ring is continuous at least in the entire 3D volume captured.

## FtsAZ filaments localize only on the mother cell side during sporulation

Next, we analyzed divisome architecture in sporulating *B. subtilis* cells. During sporulation, the division site shifts closer to a pole as opposed to medial division during vegetative growth (*Figure 1*). We again observed two series of dots in dividing sporulating cells, but remarkably, they were visible only on the mother cell side of the leading edge of the invaginating sporulation septum (*Figure 7A–C*, *Figure 7—figure supplements 1–5*). Of note, the membranes at the leading edge of the nascent septum in sporulating cells were not as defined as in vegetative cells, suggesting that they have a higher ratio of protein to membrane, as might be expected due to the presence of both transmembrane cell division proteins and sporulation-specific multipass transmembrane proteins (*Figure 7B, C*). Due to molecular crowding, it is difficult to ascertain the continuity of Z-rings during sporulation as compared to vegetative growth. In conclusion, our data suggest that FtsAZ filaments mediating cell division localize differently during vegetative growth and sporulation. During vegetative growth, they span the leading edge of the invaginating septum uniformly, whereas during sporulation they localize only on the mother cell side (*Figure 7D*).

## FtsAZ filaments tracking the division plane may dictate septal thickness

Previous data demonstrated that upon closure the polar sporulation septum is just one-fourth of the thickness of the medial vegetative septum (~25 nm vs. ~80 nm) (*Illing and Errington, 1991*; *Tocheva et al., 2013*). Recent studies suggest that FtsZ treadmilling drives the circumferential motion of septal PG synthases and condensation of the Z-ring increases recruitment of PG synthases in *B. subtilis* (*Bisson-Filho et al., 2017*; *Squyres et al., 2020*). The observation that the sporulation septum is thinner than the vegetative septum and that FtsZ filaments localize differently during the two conditions led us to hypothesize that there may be a direct correlation between the number of FtsZ filaments at the division site and the septal thickness during septal biogenesis.

To investigate this, we first measured the septal thickness in dividing sporulating and vegetative cells using cryo-FIB-ET images to determine if the sporulating septum is thinner than the vegetative septum as it is being formed or only upon closure as was previously reported (*Illing and Errington, 1991*; *Tocheva et al., 2013*; *Figure 7E*). Our data showed that the thickness of the invaginating septa in vegetative cells was ~50 ± 3 nm compared to ~80 nm when the septum formation is completed (*Figure 7E*, *Figure 7—source data 1*). This suggests that during vegetative growth either additional PG is incorporated into the septum upon its closure or the already present PG is remodeled and expands in volume after synthesis. We favor the latter possibility as during vegetative growth the septum splits to generate two equal daughter cells upon closure. This process requires the activity of several cell wall hydrolases that cleave crosslinks in septal PG, leading to the expansion of the cell wall by distributing forces across the cell surface, which would result in an increase in septal thickness (*Huang et al., 2008*; *Lee and Huang, 2013*). On the other hand, the thickness of the invaginating septa in sporulating cells was almost half of that of vegetative cells (~25 ± 2.5 nm, *Figure 7E*) during constriction. Our previous results demonstrate that the thickness of the sporulation septum upon closure is ~23 nm (*Khanna et al., 2019*), close to the observed thickness of the invaginating sporulation septa (*Figure 7E*). Hence, contrary to the vegetative septum, the sporulation septum does not thicken upon closure. This further supports the idea that the thickening of vegetative septa upon closure might be related to the expansion of the cell wall due to PG hydrolysis during cell separation. The mother cell and the forespore do not separate after septation, and therefore the cell wall hydrolases that participate in splitting of the two daughter cells during vegetative growth would not participate in the same capacity during sporulation. Our results, therefore, suggest that even at the onset of cell division the sporulating septum is thinner than the vegetative septum.

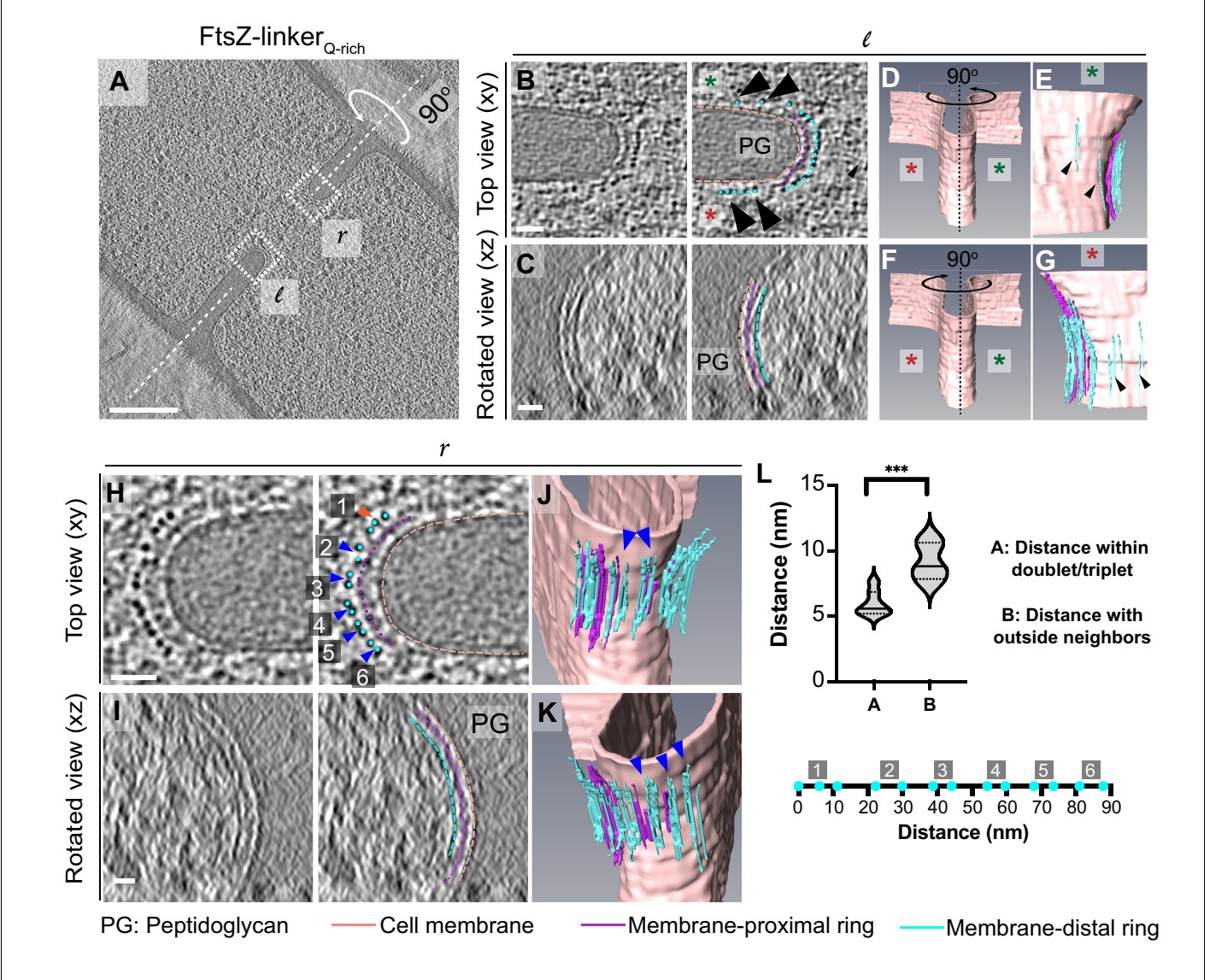

**Figure 6.** Cytoskeletal filaments in FtsZ-linker$_{Qrich}$ strain. (A) Slice through a tomogram of a dividing cell expressing FtsZ-linker$_{Qrich}$, as in *Figure 5A*. The insets (*l* for left and *r* right for right side of the septum) highlight the leading edge of the invaginating septum. (B) Left panel: zoomed-in view of the inset in (A) corresponding to '*l*' in the xy coordinate plane. Right panel: same as left with peptidoglycan (PG, gray), cell membrane (peach), membrane-proximal series of dots (pink) and membrane-distal series of dots (blue) highlighted. Same color scheme is followed throughout. Black arrows indicate membrane-distal dots that are likely not tethered to the membrane via membrane-proximal dots. Green and red stars are used to differentiate the two opposite sides of the dividing septum. (C) Left panel: view of the septal disc corresponding to (B) in the xz coordinate plane obtained by rotating the cell around its short axis by 90°. Right panel: same as left with different cellular parts and cytoskeletal filaments highlighted. (D) Segmentation of the cell membrane (peach) corresponding to (B) and (C). Red and green stars indicate the two opposite sides. (E) View of the side highlighted by the green star obtained by rotating the cell by 90° as indicated in (D). Membrane-distal dots highlighted by black arrows in the right panel of (B) are highlighted. (F) Same as (D) except that the cell is rotated along its short axis by 90° to get a view of the septum side indicated by the red star. (G) View of the side highlighted by the red star. Membrane-distal dots highlighted by black arrows in the right panel of (B) are highlighted. (H) Left panel: zoomed-in view of the inset in (A) corresponding to '*r*' in the xy coordinate plane. Right panel: same as left with different cellular parts highlighted. Doublets of the membrane-distal series of dots are indicated by blue arrows (labeled 2–6), and a possible triplet is indicated with an orange arrow (labeled 1). (I) Left panel: view of the septal disc corresponding to (H) in the xz coordinate plane obtained by rotating the cell around its short axis by 90°. Right panel: same as left with different cellular parts and cytoskeletal filaments highlighted. (J, K) Two views of the annotated cell membrane, membrane-proximal and membrane-distal filaments corresponding to (H) and (I). Doublets of membrane-distal filaments highlighted by blue arrows in the right panel of (H) are indicated in (J) and (K). (L) Violin plot showing the quantification of distances of membrane-distal dots indicated in the right panel of (H) within the doublets/triplet (as indicated by blue/orange arrows in the right panel of H) vs. distance of membrane-proximal dots with neighbor outside of the proposed doublet/triplet (***p≤0.001, unpaired t-test). Below the plot, membrane-proximal dots indicated in blue in the right panel in (H) are flattened

*Figure 6 continued on next page*

*Figure 6 continued*

and drawn to scale for context. Clusters labeled 1–6 in the right panel of (H) are also indicated. Scale bars: (A) 200 nm, (B, C, H, I) 25 nm. Scale bars are omitted from (D–G) and (J, K) owing to their perspective nature. See also *Figure 6—figure supplements 1* and *2*.

The online version of this article includes the following figure supplement(s) for figure 6:

**Figure supplement 1.** Cytoskeletal filaments in FtsZ-linker$_{Q-rich}$.

**Figure supplement 2.** Intensity of cytoskeletal filaments in FtsZ-linker$_{Q-rich}$.

Next, we investigated if the number of FtsZ filaments tracking the division plane during vegetative growth and sporulation differs. It was often not possible to count the exact number of FtsZ filaments (or the number of electron-dense puncta around the leading edge) in cryo-FIB-ET data due to low signal-to-noise ratio at the division site, especially during sporulation given the molecular crowding at the leading edge of the constricting septum. However, our data indicate that the distance between individual electron-dense puncta representing FtsZ is similar in both vegetative and sporulating septa. Hence, we measured the distance spanned by the membrane-distal band of FtsZ bundle (end-to-end distance) as a proxy for the abundance of the cytoskeletal machinery during cell division. On average, FtsZ bundles spanned twice the length in vegetative cells compared to sporulating cells (~51 ± 2.9 nm vs. ~27 ± 3.6 nm, *Figure 7F*, *Figure 7—source data 2*), indicating that there are twice as many FtsZ filaments participating in cell division during vegetative growth as compared to sporulation. Based on these data, and the fact that the dynamic property of FtsZ filaments regulates the insertion of the cell wall material during septal biogenesis (*Bisson-Filho et al., 2017*; *Squyres et al., 2020*), we propose that the sporulation septum being thinner than the vegetative septum is likely a consequence of fewer FtsZ filaments tracking the division plane during sporulation.

## SpoIIE affects the localization of FtsAZ filaments during sporulation

We next probed factors leading to the distinct pattern of FtsAZ localization during vegetative growth and sporulation. We focused on dissecting the role of SpoIIE, a sporulation-specific integral membrane protein proposed to regulate the formation of the sporulation septum for several reasons. First, SpoIIE localizes to the invaginating sporulation septum in an FtsZ-dependent manner (*Arigoni et al., 1995*; *Levin et al., 1997*; *Lucet et al., 2000*). Biochemical evidence suggests that the central FtsZ-binding domain and possibly the N-terminal transmembrane domain of SpoIIE play a role in its interaction with FtsZ (*Carniol et al., 2005*; *Lucet et al., 2000*). Second, previous electron micrographs demonstrated that certain *spoIIE* mutants form thicker polar septa compared to wild-type sporangia (*Barák and Youngman, 1996*; *Illing and Errington, 1991*). These mutations correspond to either transposon insertions or point mutations in *spoIIE* locus that do not produce any active gene product and hence represent the null phenotype. Third, recent evidence suggests that SpoIIE preferentially localizes to the forespore side of the dividing septum, creating an asymmetry during polar division that might contribute to the asymmetry of FtsZ (*Eswaramoorthy et al., 2014*; *Guberman et al., 2008*; *Wu et al., 1998*). Hence, we set out to determine if SpoIIE regulates the divisome architecture during sporulation.

We first examined divisome architecture in *spoIIE* null mutant sporangia. Surprisingly, we noted that divisome architecture in *spoIIE* is similar to that in vegetative cells, with two series of dots corresponding to FtsAZ filaments present uniformly along the leading edge of the invaginating septum, in contrast to wild-type sporangia where they were present only on the mother cell side (*Figure 8A–C, N*). We also observed several *spoIIE* sporangia with abortive septa (*Figure 8—figure supplement 1*). Also, at ~42 ± 4 nm, the invaginating sporulation septum in *spoIIE* sporangia was comparable in thickness to the invaginating vegetative septum, implicating SpoIIE in regulating both the localization of FtsAZ filaments and the septal thickness during sporulation (*Figure 8O*, *Figure 8—source data 1*). Of note, even though the length spanned by the bundle of FtsZ filaments in *spoIIE* sporulation septa was similar to that in the wild-type vegetative septa (*Figure 8—figure supplement 2A*), the invaginating *spoIIE* sporulation septa were still slightly thinner (by ~10 nm) than the invaginating vegetative septa. This suggests that there may be other unidentified factors in addition to SpoIIE that regulate the thickness of the sporulation septum.

SpoIIE has three domains: an N-terminal transmembrane domain with 10 membrane-spanning segments (region I), a central domain that promotes oligomerization of SpoIIE and its interaction

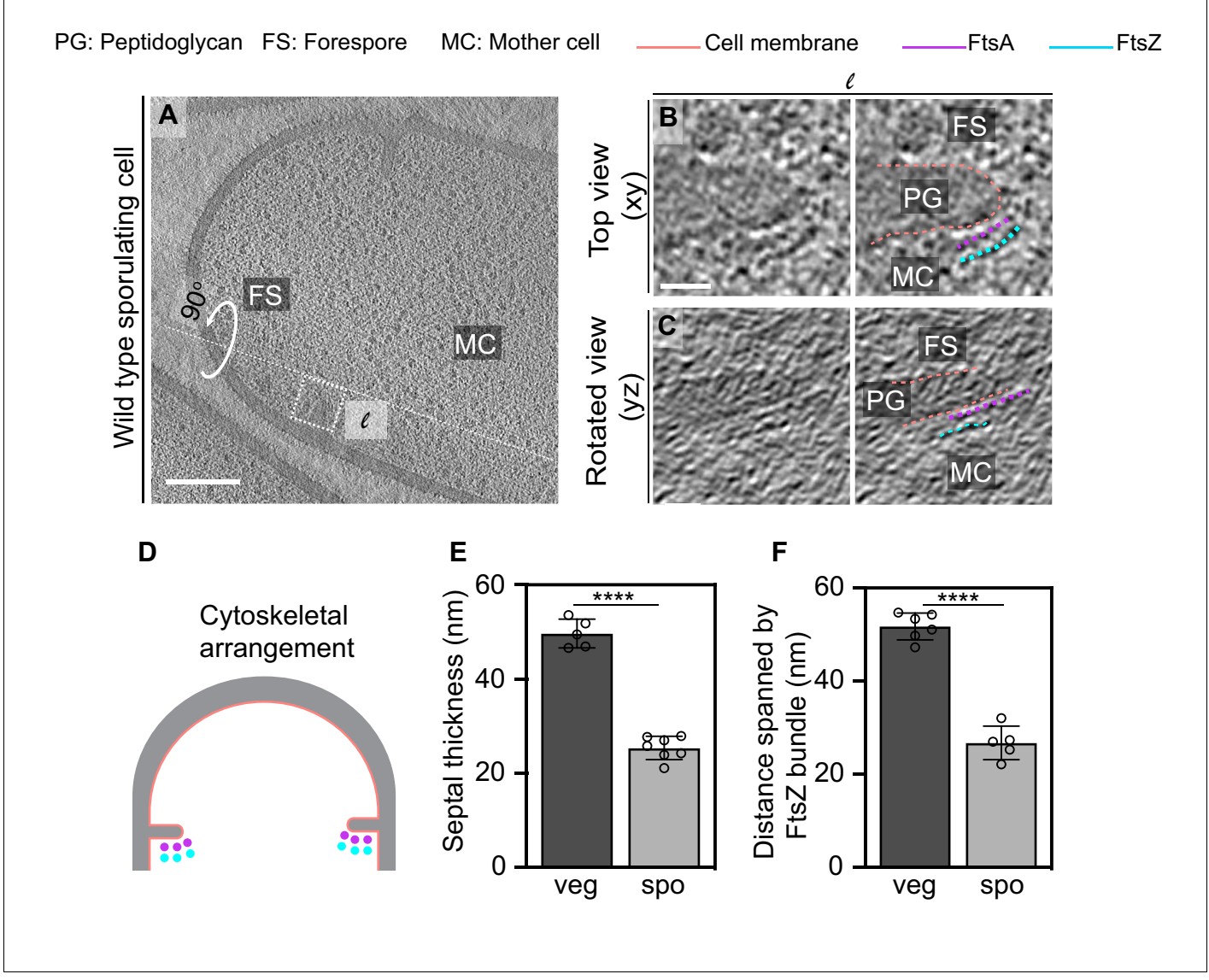

**Figure 7.** FtsAZ filaments during sporulation and septal thickness measurements. (**A**) Slice through a tomogram of a dividing sporulating cell. (**B**) Left panel: zoomed-in view of the inset in (**A**) in the xy coordinate plane. Right panel: same as left with peptidoglycan (gray), cell membrane (peach), FtsA bundle (pink) and FtsZ bundle (blue) highlighted. (**C**) Left panel: view of the septal disc corresponding to (**B**) in the yz coordinate plane obtained by rotating the cell around its long axis near the left side of the invaginating septum by 90°. Right panel: same as left with cellular parts and FtsAZ filaments highlighted. (**D**) Schematic of the arrangement of the cytoskeletal machinery during sporulation. (**E, F**) Bar graphs depicting (**E**) septal thickness and (**F**) distance spanned by FtsZ bundle in wild-type vegetative and sporulating cells. For both, error bars indicate standard deviation. Each dot indicates a sample point. (****p≤0.0001, unpaired t-test). Scale bars: (**A**) 200 nm, (**B, C**) 25 nm. See also *Figure 7—figure supplements 1–5*. The online version of this article includes the following source data and figure supplement(s) for figure 7:

**Source data 1.** Septal thickness in dividing vegetatively growing cells.

**Source data 2.** Distance spanned by FtsZ bundle in dividing vegetatively growing cells and wild-type sporulating cells.

**Figure supplement 1.** Visualizing cytoskeletal filaments in cross-sectional views in sporulating cells.

**Figure supplement 2.** Additional example of FtsAZ localization during sporulation.

**Figure supplement 3.** Additional example of FtsAZ localization during sporulation.

**Figure supplement 4.** Additional example of FtsAZ localization during sporulation.

**Figure supplement 5.** Cytoskeletal filaments in SpoIIIE[ATP-] mutant sporangia.

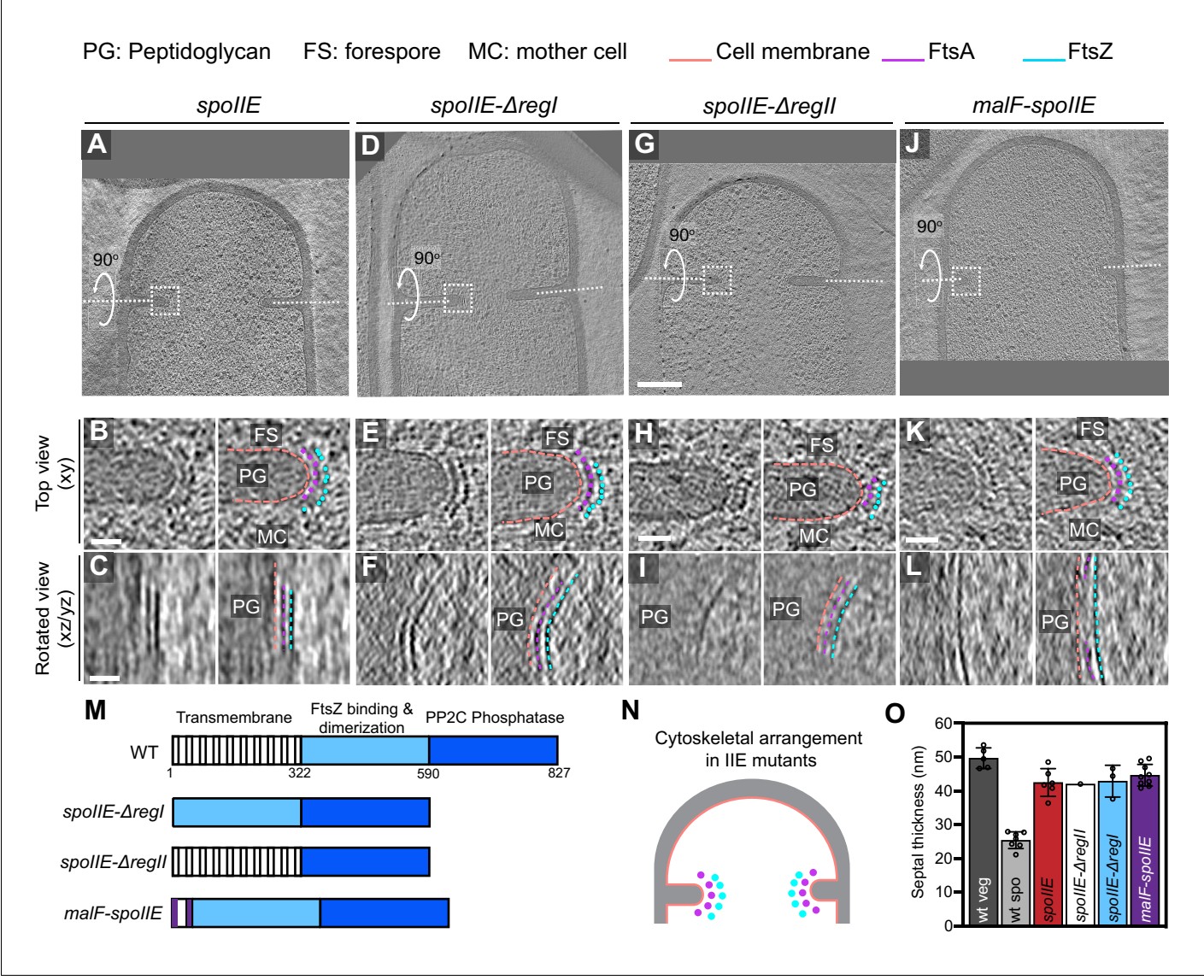

**Figure 8.** Localization of FtsAZ filaments in SpoIIE mutant sporangia. (A, D, G, J) Slice through a tomogram of a dividing (A) *spoIIE*, (D) *spoIIE-ΔregI*, (G) *spoIIE-ΔregII* and (J) *malF-spoIIE* sporangia. Insets highlight the left side of the invaginating septum for each case. (B, E, H, K) Left panel: zoomed-in view of the inset in (A, D, G, J), respectively, in the xy coordinate plane. Right panel: same as left with peptidoglycan (PG), forespore (FS) and mother cell (MC) compartments indicated in all. Cell membrane (peach), FtsA bundle (pink) and FtsZ bundle (blue) are also highlighted for all. (C, F, I, L) Left panel: view of the septal disc corresponding to (B, E, H, K), respectively, in the xz/yz coordinate plane obtained by rotating the cell around its short axis or long axis by 90° (yz plane for I and xz plane for the rest of the panels). Right panel: same as left with cellular parts and FtsAZ filaments highlighted. (M) Schematic highlighting the different domains of SpoIIE and construction of different *spoIIE* mutants. (N) Schematic of the arrangement of the cytoskeletal machinery in *spoIIE* mutant sporangia from (A) to (L). PG (gray), cell membrane (peach), FtsA dots (pink) and FtsZ dots (blue) are indicated. (O) Bar graph depicting the septal thickness in wild-type vegetative, wild-type sporulating cells and *spoIIE* mutant sporangia. Error bars indicate standard deviation. Each dot indicates a sample point. Scale bars: (A, D, G, J) 200 nm, (B, C, E, F, H, I, K, L) 25 nm. See also *Figure 8—figure supplements 1–4*, *Figure 8—source data 1*.

The online version of this article includes the following video, source data, and figure supplement(s) for figure 8:

**Source data 1.** Septal thickness in dividing vegetatively growing cells, wild-type sporulating cells and different *spoIIE* sporangia.

**Figure supplement 1.** SpoIIE sporangia with abortive septa.

**Figure supplement 2.** Characterization of SpoIIE mutant sporangia.

**Figure supplement 3.** Additional slice for SpoIIE-ΔregII sporangium shown in *Figure 8H*.

**Figure supplement 4.** Additional example of FtsAZ localization in SpoIIE-ΔregII sporangium.

**Figure 8—video 1.** Series of slices through the cryo-electron tomogram of a dividing sporulating *B. subtilis* cell shown in *Figure 8H* and *Figure 8—figure supplement 3*.

*Figure 8 continued on next page*

with FtsZ (region II) and a C-terminal phosphatase domain that is involved in the activation of the forespore-specific transcription factor σ^F after septum formation (region III) (*Carniol et al., 2005*). Since regions I and II have been implicated in regulating the interaction of SpoIIE with FtsZ (*Carniol et al., 2005*; *Lucet et al., 2000*), we acquired cryo-FIB-ET images of the following *spoIIE* mutant strains: (1) deletion of region I (or *spoIIE-ΔregI*), (2) deletion of region II (or *spoIIE-ΔregII*), and (3) replacement of 10 membrane-spanning segments of region I by 2 membrane-spanning segments from *E. coli* MalF (or *malF-spoIIE*). These strains also had a deletion in the *spoIIA* operon to uncouple the role of SpoIIE in polar division from its role in σ^F activation (*Figure 8—figure supplement 2B, C*). Our cryo-FIB-ET data demonstrated that the organization of FtsAZ filaments at the polar septum and the septal thickness in all the three mutant strains is similar to *spoIIE* sporangia (*Figure 8D–O*, *Figure 8—figure supplements 3* and *4*, *Figure 8—video 1*). These data indicate that both the transmembrane and the FtsZ-binding domains of SpoIIE are essential to regulate the localization of FtsAZ filaments and septal thickness during sporulation.

## SpoIIE affects the localization of FtsA on the forespore side of the septum

SpoIIE is present exclusively on the forespore side of the septum even before the onset of membrane constriction (*Eswaramoorthy et al., 2014*; *Guberman et al., 2008*; *Wu et al., 1998*), and our data show that both the membrane-spanning and the FtsZ-binding domains of SpoIIE mediate differential localization of FtsAZ filaments during sporulation. Since SpoIIE directly binds FtsZ, one possible way SpoIIE could affect the localization of FtsZ filaments on the forespore side would be by competing directly with other cell division proteins that bind to FtsZ, including FtsA. Hence, we hypothesized that the interaction of SpoIIE with FtsZ may disrupt the interaction of FtsA with FtsZ on the forespore side. This could either be a result of direct competition faced by FtsA from SpoIIE for the binding site on FtsZ or due to possible membrane curving on the forespore side mediated by SpoIIE oligomers upon insertion of their large membrane-spanning segment, as has been previously reported in protein-packed environments (*Stachowiak et al., 2012*). In support of this, using time-lapse fluorescence microscopy, in a few cells, we observed two foci on each side of the cell at the division septum corresponding to fluorescently tagged FtsA, a bright signal (orange arrow, *Figure 9A*) that constricts with the invaginating membrane and a faint signal (yellow arrow, *Figure 9A*) that remains at the intersection of the lateral and the septal cell wall above the brighter signal (likely on the forespore side). We also used structured illumination microscopy (SIM) to observe FtsA-mNeonGreen in dividing sporulating cells at a higher resolution. Again, in a few cells, we observed bright FtsA puncta (orange arrow, *Figure 9B, C*) that colocalize with the invaginating septum and a faint FtsA punctum (yellow arrow, *Figure 9B, C*) that is slightly above the brighter signal, presumably on the forespore side. Due to the resolution of the optical microscopy techniques, we cannot state with confidence that the fainter FtsA signal is in the forespore, but our data suggest that there are some FtsA filaments that remain at the intersection of the lateral and the septal cell wall above the FtsA filaments that constrict with the invaginating membrane. It is possible that SpoIIE causes the disassociation of these residual FtsA filaments from the divisome in the forespore as SpoIIE is only present on the forespore side of the invaginating septum so that cytokinesis exclusively proceeds from the mother cell side (*Figure 9D*). Further experiments will be required to conclusively establish the mechanism by which SpoIIE interacts with different components of the divisome to regulate cell division during sporulation.

## Discussion

In this study, we elucidate the differences in the architecture of the cell division machinery in *B. subtilis* during vegetative growth and sporulation and their impact on septal PG thickness. First, we show that during vegetative growth, two cytoskeletal rings localize uniformly around the leading edge of the invaginating medial septum to orchestrate cell division, a membrane-proximal ring of FtsA

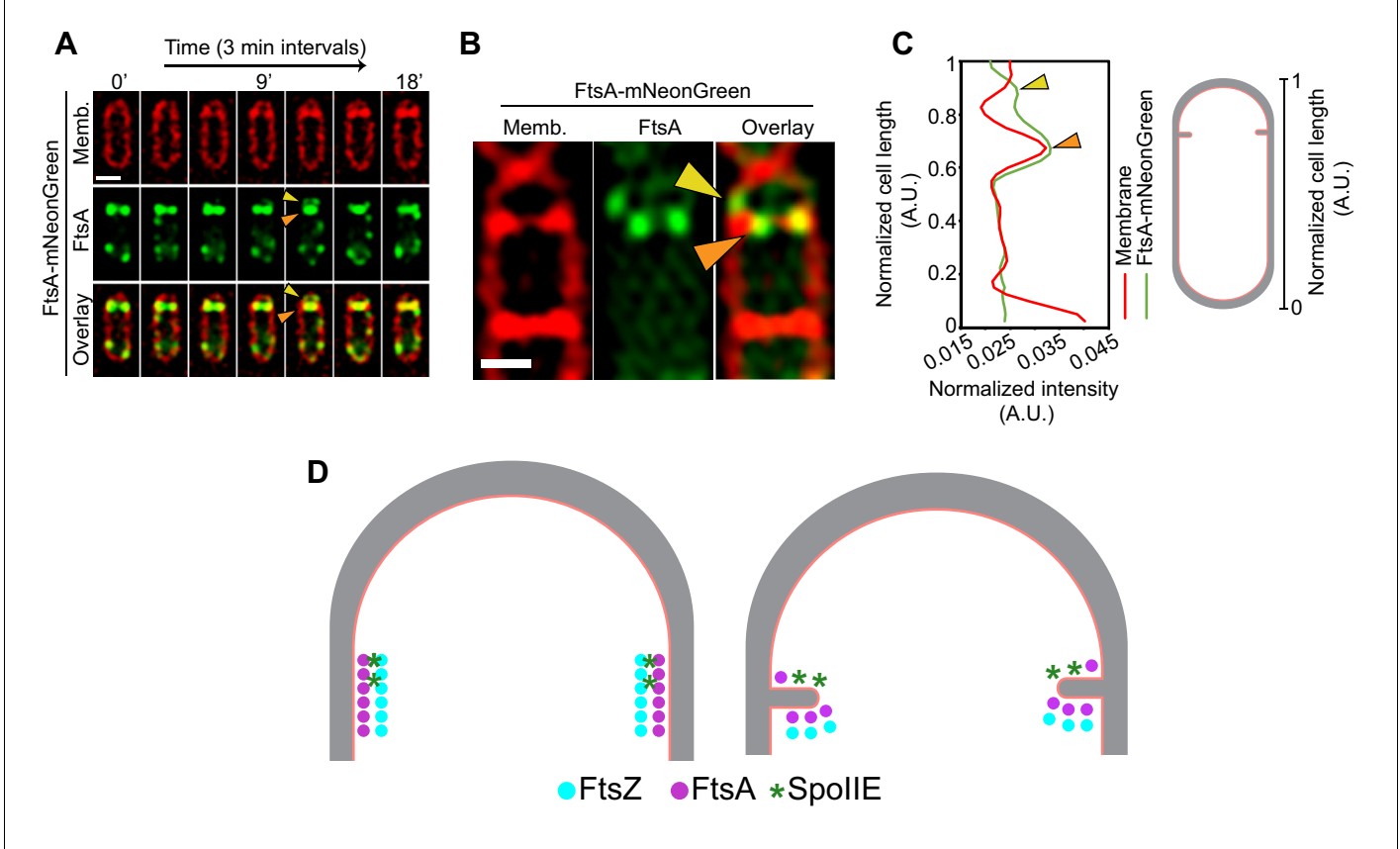

**Figure 9.** Role of SpoIIE in modulating FtsAZ filament localization. (**A**) Time-lapse fluorescence microscopy of a dividing sporangium with FtsA in green and membranes in red. Images are taken every 3 min. Between 12 and 15 min, two FtsA rings are visible, one that constricts and the other that seemingly stays behind at the forespore side of the septum. Yellow arrows indicate FtsA on the forespore side that does not constrict. (**B**) Structured-illumination microscopy of a dividing sporangia with FtsA in green and membranes in red. Puncta corresponding to FtsA are more clearly visible, two (orange) on either side of the septum that constrict and one (yellow arrow) that remains above the invaginating septum near the lateral edge, presumably on the forespore side. (**C**) Line graph showing the normalized intensity of the membrane (red) and FtsA-mNeonGreen (green) signal along the normalized length of the cell. Two peaks corresponding to two puncta of FtsA in (**B**) are indicated by orange and yellow arrows, respectively. (**D**) A possible model of how SpoIIE (green asterisks) affects the localization of FtsA (pink) and FtsZ (blue) filaments during sporulation (top view of the cell shown). Peptidoglycan (gray) and cell membrane (peach) are also highlighted. SpoIIE molecules are preferentially present on the forespore side and hence bind to FtsZ on the forespore side (brown asterisks). This may prevent binding of FtsA filaments (and likely other cell division proteins) to FtsZ on the forespore side and they stay behind at the forespore edge, while FtsAZ filaments on the mother cell side constrict. Scale bars: (**A, B**) 1 μm. See also *Figure 9—figure supplement 1*.

The online version of this article includes the following figure supplement(s) for figure 9:

**Figure supplement 1.** FtsAZ in the forespore during engulfment.

filaments that tethers the membrane-distal ring of FtsZ filaments to the membrane (*Figures 3– 6*). Second, we demonstrate that during the asymmetrically positioned division event in the cell at the onset of sporulation, FtsAZ filaments are assembled only on the mother cell side of the invaginating polar septum (*Figure 7A–D*), demonstrating a surprising degree of asymmetry even in the localization of the division machinery. Third, our data suggest that during septal biogenesis the number of FtsAZ filaments tracking the sporulation septum is approximately half of that in the vegetative septum, which likely gives rise to a thinner sporulation septum by mediating the circumferential motion of fewer PG synthases around the division plane during sporulation (*Figure 7E, F*). Finally, we show that a sporulation-specific protein SpoIIE, which is essential for cell-specific gene expression after polar septation, is also essential for the asymmetric assembly of FtsAZ filaments and septal thickness at the onset of sporulation (*Figures 8* and *9*). These results demonstrate that the

sporulation septum is both asymmetrically positioned in the cell and that it is assembled by an asymmetrically localized division machinery.

Most of our understanding of bacterial cytokinesis comes from studies in model Gram-positive and Gram-negative bacteria, namely *B. subtilis* and *E. coli*, respectively. Although the overall mechanism of cytokinesis is conserved in that a Z-ring that recruits other components of the divisome is formed at the division site, there are subtle differences with regards to the spatial and the temporal regulation of the process and the composition of the divisome for the two types of bacteria. Previously, only a single cytoskeleton ring of FtsZ was observed using cryo-ET in Gram-negative bacteria including *E. coli* and *C. crescentus* (*Li et al., 2007*; *Szwedziak et al., 2014*). However, we observed two rings corresponding to FtsA and FtsZ filaments in cryo-FIB-ET images of *B. subtilis* during both vegetative growth and sporulation, even though the ratio of FtsZ to FtsA is same (5:1) in *B. subtilis* and *E. coli* and there are enough molecules of FtsA to form a circumferential A-ring in both species (*Feucht et al., 2001*; *Rueda et al., 2003*). But an A-ring is apparent only in *B. subtilis,* although it appears more patchy as compared to the mostly continuous Z-ring. One possible explanation for this difference between species is that cell division in *B. subtilis* involves the constriction of a septal disc composed of thicker septal PG (~25 nm during sporulation and ~50 nm during vegetative growth) than in *E. coli* or *C. crescentus*, where the septal disc consists of only ~4-nm-thick PG. Hence, cytokinesis in *B. subtilis* may require more FtsA filaments in order to tether the increased number of PG synthetases that are required to build a thicker septum during constriction. Other differences between our data and that from Gram-negative bacteria are the decreased lateral spacing between FtsZ filaments (~5.5 nm in *B. subtilis* vs. 6.5–8 nm in *E. coli* and *C. crescentus*), and the shorter distances of the Z-ring and the A-ring from the membrane (~14 nm and ~6.5 nm, respectively, in *B. subtilis* vs. ~16 nm and ~8 nm, respectively, in *E. coli*). It is possible that a tighter bundling of FtsAZ filaments with each other and with the membrane in *B. subtilis* may be necessary for the synthesis of the thick septal PG. Further studies will determine if these mechanisms of Z-ring and A-ring formation are also conserved in other sporulating and non-sporulating Gram-positive bacteria.

Our result that the Z-ring is mostly continuous during vegetative growth is similar to a previous cryo-ET study that also revealed the continuity of Z-rings in both *E. coli* and *C. crescentus* (*Szwedziak et al., 2014*). However, this contrasts with super-resolution microscopy in which the Z-ring in both *B. subtilis* and *E. coli* appeared patchy with regions of low fluorescence intensity that may indicate gaps in the ring (*Holden et al., 2014*; *Strauss et al., 2012*). We note that cryo-ET has better spatial resolution than super-resolution microscopy and enables visualization of cellular structures in a near-native state. It is possible that we are unable to detect gaps in our cryo-FIB-ET data because they are either present in the cellular region that was ablated by cryo-FIB milling or not sampled due to missing wedge associated with cryo-ET data collection (*Figure 2—figure supplement 1*). However, these scenarios seem less likely because different tomograms capture different parts of the septal disc during constriction, and the ring appeared continuous in each of them. Thus, it seems likely that at least under the conditions used here the Z-ring is largely contiguous during constriction, although further studies are required to determine if this varies under other experimental conditions or at specific stages of FtsZ ring assembly, constriction or disassembly. During sporulation, we notice that the septal membranes are not as sharply defined as during vegetative growth, probably due to molecular crowding by membrane proteins that participate in cell division as well as those that are produced specifically during sporulation. Also, since FtsAZ are only present on the mother cell side and the distance spanned by the bundle around the invaginating septum is much shorter during sporulation, the filaments are detected in only a few slices in cross-sectional views as opposed to vegetative growth. These limitations make it difficult to conclude with confidence whether the Z-ring is also continuous during sporulation. However, the Z-ring is continuous during sporulation in SpoIIE sporangia in the tomograms we acquired, suggesting their continuous nature in at least under this condition during sporulation.

Our data demonstrate that during sporulation not only is there asymmetry associated with the positioning of the division septum within the cell, but also with the positioning of the division machinery at the septum, since FtsAZ filaments are only present on the mother cell side of the invaginating septum. This is an unanticipated observation. Further, we noted that only half the number of filaments assemble at the septum and track the division plane during sporulation than during vegetative growth, and that the septum is also half as thick in sporulating cells. These results suggest that

the number of FtsAZ filaments could directly dictate the number of complexes involved in PG synthesis.

A thinner septum could be beneficial during sporulation in two ways. First, several protein complexes are proposed to form channels across both septal membranes during sporulation, including SpoIIQ-SpoIIIAH and SpoIIIE (*Blaylock et al., 2004*; *Fleming et al., 2010*; *Meisner et al., 2008*; *Riley et al., 2021a*; *Yen Shin et al., 2015*). A thinner septum could facilitate interaction between the subunits located in the forespore and those in the mother cell. Second, a thinner septum may facilitate the phagocytosis-like process of engulfment, during which migration of the mother cell membrane is driven by the remodeling of the junction between the septal PG and the lateral PG via the coordinated synthesis and degradation of the septal PG (*Khanna et al., 2019*; *Ojkic et al., 2016*). This remodeling event may be more straightforward if the sporulation septum contained relatively few or potentially only one layer of PG.

In addition to facilitating the formation of a thinner septum, asymmetric distribution of FtsAZ to the mother cell side of the septum may also be beneficial for other downstream events during sporulation. First, it is possible that such a strategy allows enough FtsAZ complexes to remain in the mother cell to support a second division site at the opposite cell pole as a backup should the first septum be misassembled or fail to support the onset of cell-specific gene expression. This second polar division site is dissolved after the onset of cell-specific gene expression by either the PG degradation complex SpoIIDMP (*Abanes-De Mello et al., 2002*; *Eichenberger et al., 2001*; *Pogliano et al., 1999*) or MciZ (*Bisson-Filho et al., 2015*; *Handler et al., 2008*) in the mother cell. Retaining FtsAZ complexes in the mother cell would ensure their availability for the second division event at the distal pole.

Second, if FtsAZ complexes were present on the forespore side of the septum after septation, they may induce aberrant cell division events in the small forespore if their activity was not efficiently inhibited by the Min and the nucleoid occlusion proteins (*Marston et al., 1998*; *Wu and Errington, 2004*). Thus, ensuring that the forespore does not harbor active cytoskeletal filaments upon septation could provide an additional safety mechanism. Time-lapse microscopy of FtsA-mNeonGreen and FtsZ-mNeonGreen indicated that just after polar septation there was some FtsAZ signal in the forespore that gradually diminished as engulfment proceeded in these cells (*Figure 9—figure supplement 1*). It is possible that there are protein(s) in the forespore that mediate degradation of FtsAZ polymers that do not participate in cell division yet remain in the forespore, ensuring an extra layer of safety mechanism to prevent any aberrant cell division events in the forespore. A recent study from the lab showed that certain proteins involved in central carbon metabolism and metabolic precursor synthesis were depleted specifically in the forespore after polar septation (*Riley et al., 2021b*). Further studies will address if FtsAZ are depleted from the forespore via a similar mechanism.

Finally, we have previously shown that PG synthases, including Pbp2B, a component of the divisome, preferentially localize to the forespore side of the leading edge of the engulfing membrane (*Khanna et al., 2019*; *Ojkic et al., 2016*), where they act together with the mother cell PG degradation complex SpoIIDMP to mediate the coordinated synthesis and degradation of the cell wall to drive engulfment membrane migration (*Ojkic et al., 2016*). The mechanism by which these proteins localize to the region of the forespore membrane that is adjacent to the migrating mother cell membrane remains unclear, but our fluorescence microscopy data indicate that at the onset of membrane constriction, a punctum of FtsA-mNeonGreen remains at the corner formed by the septum and the outer envelope likely in the forespore. It is possible that other cell division proteins such as those involved in cell wall synthesis also remain at this site, where they would be poised to participate in engulfment after the completion of cell division. Development of cryo-ET compatible electron-dense tags to track the localization of individual divisome proteins in conjunction with single-molecule studies will aid in testing these hypotheses in the future.

Asymmetric cell division is a common developmental theme in eukaryotes to generate progeny with diverse cellular fates (*Li, 2013*). In these cases of asymmetric cell division, the general theme is the establishment of cell division machinery along an axis of polarity and distribution of cell fate determinants in a polarized manner along this axis. In case of *B. subtilis* sporulation, an axis of polarity is likely established when the division plane shifts from medial to polar position and cell fate determinants are differentially distributed between the future forespore and the mother cell compartments at the onset of membrane constriction. In this study, we have identified SpoIIE as a cell

fate determinant that also restricts FtsAZ filaments to the mother cell side of the asymmetrically positioned septum, likely by its preferential localization to the forespore side of the invaginating septum (*Eswaramoorthy et al., 2014*; *Guberman et al., 2008*; *Wu et al., 1998*). We show that regions I and II of SpoIIE corresponding to the transmembrane domain and the FtsZ-binding domain, respectively, regulate the septal thickness and the distribution of FtsAZ at the sporulation septum. Region III, C-terminal part of SpoIIE, consists of a phosphatase domain that is essential to activate $\sigma^F$ in the forespore by dephosphorylating SpoIIAA after septum closure. It was previously shown that a SpoIIE mutant that is enzymatically inactive in dephosphorylating SpoIIA (D746A substitution) behaved similar to wild type with respect to polar septation during sporulation (*Carniol et al., 2005*). In another study, missense mutations *spoIIE64* and *spoIIE71* that map to region III of SpoIIE and affect its phosphatase activity were shown to have thin polar septa in thin-section electron microscopy images (*Barák and Youngman, 1996*), suggesting that the phosphatase activity of SpoIIE likely does not affect the septal thickness. However, we cannot exclude the possibility that the deletion of the entire region III or some other mutations in region III may play a role in regulating the septal thickness and/or FtsAZ localization at the septum. Also, investigating the physiological relevance of septal thickness during sporulation is complicated by the pleiotropic nature of SpoIIE mutations. SpoIIE directly participates in polar septation, but it is also required for the activation of $\sigma^F$ in the forespore, and consequently, the rest of the sporulation-specific $\sigma$ factors. Although the two roles seem to depend on different parts of the protein, separating them is not straightforward, as making mutations that affect polar septation may also affect the localization of SpoIIE and, therefore, the protein might not be released to the forespore to activate $\sigma^F$. Future studies to examine more mutations in different regions of SpoIIE and their overexpression will provide better insights into the mechanism of regulation of septal thickness by SpoIIE. Further biochemical and genetic studies are required to elucidate the cellular mechanism by which the nascent sporulation septum is assembled asymmetrically, so that SpoIIE, which is required for forespore-specific gene expression, assembles on the forespore side of the nascent septum, while FtsAZ rings assembles on the mother cell face of the nascent septum.

Our data provide a significant advancement in the understanding of the organization of the divisome, its regulators and its role in septal PG synthesis in *B. subtilis*, providing, to the best of our knowledge, the first evidence for the asymmetric assembly of the divisome in bacteria and showing that this depends on the SpoIIE protein that is required for the onset of cell-specific gene expression. Future efforts to identify the molecular arrangement of FtsZ and its regulators inside the cell such as SpoIIE will aid in the development of new antimicrobials targeting the cell division machinery in important pathogens.

# Materials and methods

## Key resources table

| Reagent type (species) or resource | Designation | Source or reference | Identifiers | Additional information |
|---|---|---|---|---|
| Strain, strain background (*Bacillus subtilis* PY79) | PY79 | *Youngman et al., 1984* | Tax. ID:1415167 | Wild type |
| Strain, strain background (*Bacillus subtilis* PY79) | KK240 | This study | | *ftsZ-linker$_{Q-rich}$Ωkan* |
| Strain, strain background (*Bacillus subtilis* PY79) | KP69 | *Sandman et al., 1987* | | *spoIIE::Tn917* |
| Strain, strain background (*Bacillus subtilis* PY79) | KC548 | *Carniol et al., 2005* | | *spoIIE::phleo spoIIA::cat amyE::spoIIE-ΔregII-gfp cat spc* |
| Strain, strain background (*Bacillus subtilis* PY79) | KC549 | *Carniol et al., 2005* | | *spoIIE::phleo spoIIA::cat amyE::spoIIE-ΔregI-gfp cat spc* |
| Strain, strain background (*Bacillus subtilis* PY79) | KC538 | *Carniol et al., 2005* | | *spoIIE::phleo spoIIA::spec amyE::malF-spoIIE-ΔregI gfp spc kan* |

*Continued on next page*

*Continued*

| Reagent type (species) or resource | Designation | Source or reference | Identifiers | Additional information |
|---|---|---|---|---|
| Strain, strain background (*Bacillus subtilis* PY79) | bAB167 | *Bisson-Filho et al., 2017* | | *ftsA-mNeonGreen(SW) ftsA* |
| Strain, strain background (*Bacillus subtilis* PY79) | bAB185 | *Bisson-Filho et al., 2017* | | *mNeonGreen ftsZ* |
| Recombinant DNA reagent | pJLG142 | This study | | *ftsZ-ssrAΩloxPKmloxP* |
| Chemical compound, drug | FM4-64 | Thermo Fisher Scientific | Cat#T13320 | |
| Software, algorithm | IMOD | *Mastronarde, 1997* | http://bio3d.colorado.edu/imod/; RRID:SCR_003297 | |
| Software, algorithm | TomoSegMemTV | *Martinez-Sanchez et al., 2014* | https://sites.google.com/site/3demimageprocessing/tomosegmemtv | |
| Software, algorithm | Amira | Commercial software by Thermo Scientific (formerly FEI) | https://www.fei.com/software/amira-3d-for-life-sciences/; RRID:SCR_014305 | |
| Software, algorithm | SerialEM | *Mastronarde, 2005* | http://bio3d.colorado.edu/SerialEM/ | |
| Software, algorithm | MATLAB code to calculate intensities of cytoskeletal rings | This paper; *Source code 1* | | |
| Software, algorithm | MATLAB code to calculate GFP/mNeonGreen intensity along cell length | This paper; *Source code 2* | | |

## Strain and culture conditions

*B. subtilis* PY79 background was used for all strain constructions. A list of strains used in the study is provided in the Key resources table. All the strains were routinely grown in LB plates at 30℃ overnight. For sample preparation, cells were either grown in LB media at 30℃ to $OD_{600}$ ~0.5 for vegetative growth or by growing in ¼ diluted LB to $OD_{600}$ ~0.5–0.8 and then resuspending in A + B media at 37℃ for inducing sporulation. For wild-type sporangia, samples were prepared at an early enough time point of ~T1.5–1.75 hr after inducing sporulation to capture a mixed population of dividing cells undergoing the formation of polar septum (sporulating cells) or medial septum (vegetative cells). For other mutant sporangia, samples were also collected at ~T1.5–1.75 hr after inducing sporulation.

## Construction of KK240

Constructed by transformation of pKK238 (*ftsZ-linker$_{Q-rich}$Ωkan*) into PY79. pKK238 was constructed by removing ssrA tag of pJLG142 and incorporating Q-rich linker from FtsN of *E. coli* at the C-terminus of FtsZ.

FtsZ-linkerQ-rich sequence: (Q-rich linker region highlighted in red).

MLEFETNIDGLASIKVIGVGGGGNNAVNRMIENEVQGVEYIAVNTDAQALNLSKAEVKMQIGAKLTRGLGAGA
NPEVGKKAAEESKEQIEEALKGADMVFVTAGMGGGTGTGAAPVIAQIAKDLGALTVGVVTRPFTFEGRK
RQLQAAGGISAMKEAVDTLIVIPNDRILEIVDKNTPMLEAFREADNVLRQGVQGISDLIATPGLINLDFADVKTI
MSNKGSALMGIGIATGENRAAEAAKKAISSPLLEAAIDGAQGVLMNITGGTNLSLYEVQEAADIVASASDQD
VNMIFGSVINENLKDEIVVTVIATGFIEQEKDVTKPQRPSLNQSIKTHNQSVPKREPKREEPQQQNTVSRHTSQPA
RQQPTQLVEVPWNEQTPEQRQQTLQRQRQAQQLAEQQRLAQQSRTTEQSWQQQTRTSQAAPVQAQPRQSKPA
SSQQPYQDLLQTPAHTTAQSKPQQD.

## Cryo-FIB-ET workflow

QUANTIFOIL R2/1 200 mesh holey carbon Cu grids (Quantifoil Micro Tools) were glow discharged using PELCO easiGlow (Ted Pella). 7–8 μl of diluted liquid culture was deposited onto the grids,

which were mounted to a manual plunger (Max Planck Institute of Biochemistry) and manually blotted using Whatman No. 1 filter paper for 3–4 s from the side opposite to where the cells were deposited such that the cells formed a homogeneous monolayer on the grid surface. The grids were then plunge-frozen in a 50–50 mixture of liquid ethane and propane (Airgas) cooled to liquid nitrogen temperature and subsequently clipped onto Cryo-FIB Autogrids (Thermo Fisher Scientific). All subsequent transfers were performed in liquid nitrogen.

Frozen-hydrated cells were micromachined either inside Scios or Aquilos DualBeam instruments each equipped with a cryo-stage (Thermo Fisher Scientific). An integrated gas injection system (GIS) was used to deposit an organometallic platinum layer inside the FIB chamber to protect the specimen surface and avoid uneven thinning of cells. In case of specimens prepared in Aquilos, the specimen was also sputter coated with inorganic platinum layer prior to GIS deposition to prevent charging during imaging. FIB milling was then performed using two rectangular milling patterns to ablate the top and the bottom parts of the cells in steps of decreasing ion beam currents at a nominal tilt of 11°–15°, which translates into a milling angle of 4°–8°. Initial rough milling was performed using ion beam currents of 0.5–0.3 nA followed by intermediate milling at 0.1 nA or 50 pA. Finally, fine milling was conducted at 30 pA or 10 pA. In each of the steps, the distance between the two rectangular milling patterns was sequentially decreased to get ~100–300-nm-thick lamellas.

Tilt series were collected using SerialEM software (*Mastronarde, 2005*) either in 300-keV Tecnai G2 Polara or in 300-keV Titan Krios (Thermo Fisher Scientific), both equipped with a Quantum post-column energy filter (Gatan) and a K2 Summit 4k × 4k pixel direct detector camera (Gatan). The TEM magnification corresponded to a camera pixel size of either 0.612 nm (for data acquired on Polara) or 0.534 nm or 0.426 nm (for data acquired on Krios). The tilt series were usually collected from −61° to + 61° depending on the quality of the specimen with an increment ranging from 2° to 3° with a defocus of −5 µm following either the bidirectional or the dose-symmetric tilt scheme in low-dose mode. For both schemes, the zero of the tilt series was defined by taking into account the pre-tilt of the lamella. The K2 detector was operated in counting mode and images divided into frames of 0.1 s. The cumulative dose for each tilt series was ~50–150 e⁻/Å$^2$.

## Tomogram reconstruction and segmentation

MotionCor2 (*Zheng et al., 2017*) was used to align the frames and dose-weigh the tilt series according to the cumulative dose. Subsequent alignment of the tilt series was done in IMOD (*Kremer et al., 1996*) using patch-tracking in the absence of fiducials and the tomograms reconstructed using weighted back-projection method. For purposes of representation and segmentation, tomograms were binned 3× or 4×. Semi-automatic segmentation of the membranes was performed using TomosegmemTV (*Martinez-Sanchez et al., 2014*) followed by manual refinement in Amira software package (Thermo Fisher Scientific). FtsAZ filaments were manually traced in Amira.

## Image analysis
### Distribution of intensities of cytoskeletal rings

For *Figure 3—figure supplement 2* and *Figure 6—figure supplement 2*, the membrane-proximal and the membrane-distal cytoskeletal rings were masked using Adobe Photoshop and the corresponding intensity values for each pixel were extracted using a custom-built MATLAB script (*Source code 1*). The distribution of these values for the two rings was depicted as box-and-whisker plot alongside those corresponding to cell membrane, cytoplasm and cell wall (PG) that were used as controls.

### Distances between cytoskeletal rings and membrane

For *Figure 5B* (*Figure 5—source data 1*), a medial slice corresponding to the orthogonal view (xz) of the respective tomograms was taken from the z-stack wherein the cytoskeletal rings were clearly visible. The distances of the ring from the cell membrane were then calculated using the Fiji plugin 'points to distance'.

### Distance spanned by FtsZ bundle

For *Figure 7F,* a medial slice corresponding to the top view (xy) of the respective tomograms was taken from the Z-stack wherein the FtsZ bundle was clearly visible. The distance spanned by the FtsZ bundle along the curve of the membrane was then measured in Fiji.

### Fluorescence microscopy for batch cultures

Approximately 12 µl of samples were taken at indicated time points and transferred to 1.2% agarose pads that were prepared using either sporulation resuspension media (for sporulating cultures) or ¼ LB media (for vegetative growth). Membranes were stained with 0.5 µg/ml of FM4-64 (Thermo Fisher Scientific) that was added directly to the pads. An Applied Precision DV Elite optical sectioning microscope equipped with a Photometrics CoolSNAP-HQ2 camera was used to visualize the cells. The images were deconvolved using SoftWoRx v5.5.1 (Applied Precision). For all fluorescence images, the medial focal plane of the image is shown. Excitation/emission filters were TRITC/CY5 for membrane imaging and FITC/FITC to visualize GFP or mNeonGreen signal.

### Calculating GFP intensity

For *Figure 8—figure supplement 2C*, ~15–16 cells with clear evidence of septal biogenesis were manually selected for each strain and all the images aligned such that the polar septum lies in the same orientation for all cells as shown in the inset of *Figure 8—figure supplement 2Ci*. To get the linear profile of GFP intensity for each cell, the data was grouped into smaller bins of approximately equal area using a custom-built MATLAB script (*Source code 2*). Normalized GFP intensities were then plotted for each cell along its normalized length as depicted by different curves in the graphs.

### 3D-structured illumination microscopy (3D-SIM)

For *Figure 9B*, cells were grown as indicated above and stained with 0.5 µg/ml FM4-64. An Applied Precision/GE OMX V2.2 Microscope was then used to image them. Raw data were sequentially taken by SI-super-resolution light path to collect 1.5-mm-thick specimens in 125 nm increments in the z-axis with compatible immersion oils (Applied Precision). Standard OMX SI reconstruction parameters were then used in DeltaVision SoftWoRx Image Analysis Program to reconstruct the images. To plot the graph in *Figure 9C*, the data corresponding to membrane and mNeonGreen intensity was grouped into smaller bins of approximately equal area along the normalized length of the cell (as indicated in the inset) using a custom-built MATLAB script (*Source code 2*).

### Spore titer assay

Strains were grown in triplicates in 2 ml of DSM media for 24 hr at 37°C followed by heating at 80°C for 20 min. Then, serial dilutions for each strain were prepared and spotted on LB plates. Number of colonies were then used as a marker to calculate spore titers.

## Acknowledgements

This work was supported by the National Institutes of Health R01-GM057045 (to KP and EV) and the National Science Foundation MRI grant NSF DBI 1920374 (to EV). We acknowledge the use of the UC San Diego cryo-EM facility, which was built and equipped with funds from UC San Diego and an initial gift from Agouron Institute, and of the San Diego Nanotechnology Infrastructure (SDNI) of UC San Diego, a member of the National Nanotechnology Coordinated Infrastructure, supported by the NSF grant ECCS-1542148. We thank Dr. Marcella Erb for help with 3D-SIM experiments. We thank Richard Losick, Kumaran Ramamurthi and Ethan Garner for the gift of strains. We thank Joe Pogliano and other members of the Pogliano and Villa labs for their many helpful comments on the manuscript.

## Additional information

### Funding

| Funder | Grant reference number | Author |
|--------|------------------------|--------|
| National Institutes of Health | R01-GM057045 | Kit Pogliano Elizabeth Villa |
| National Science Foundation | DBI 1920374 | Elizabeth Villa |

The funders had no role in study design, data collection and interpretation, or the decision to submit the work for publication.

### Author contributions

Kanika Khanna, Conceptualization, Formal analysis, Investigation, Methodology, Writing - original draft; Javier Lopez-Garrido, Conceptualization, Writing - review and editing; Joseph Sugie, Formal analysis, Writing - review and editing; Kit Pogliano, Elizabeth Villa, Conceptualization, Supervision, Funding acquisition, Writing - review and editing

### Author ORCIDs

Kanika Khanna  https://orcid.org/0000-0001-7150-0350
Javier  Lopez-Garrido  https://orcid.org/0000-0002-8907-502X
Joseph Sugie  http://orcid.org/0000-0003-2911-1807
Kit Pogliano  https://orcid.org/0000-0002-7868-3345
Elizabeth Villa  https://orcid.org/0000-0003-4677-9809

### Decision letter and Author response

Decision letter https://doi.org/10.7554/eLife.62204.sa1
Author response https://doi.org/10.7554/eLife.62204.sa2

## Additional files

### Supplementary files

• Source code 1. Custom-built MATLAB script to calculate the intensities of cytoskeletal rings from cryo-ET data (see *Figure 3—figure supplement 2* and *Figure 6—figure supplement 2*).

• Source code 2. Custom-built MATLAB script to calculate normalized GFP/mNeonGreen/FM$-64 intensities for each cell along its normalized length (see *Figure 8—figure supplement 2C* and *Figure 9C*).

• Transparent reporting form

### Data availability

All data generated or analysed during this study are included in the manuscript and supporting files. Additionally, we have deposited representative tomograms for each growth condition in the Electron Microscopy Data Bank (EMDB) accession codes EMD-23963, EMD-23964, EMD-23965, EMD-23966, EMD-23967 and EMD-23968, and tilt series in the Electron Microscopy Public Image Archive (EMPIAR) database accession code EMPIAR-10710.

The following datasets were generated:

| Author(s) | Year | Dataset title | Dataset URL | Database and Identifier |
|-----------|------|---------------|-------------|-------------------------|
| Khanna K, Lopez-Garrido J, Sugie J, Pogliano K, Villa E | 2021 | Tomogram of a dividing vegetative cell of *Bacillus subtilis* | https://www.ebi.ac.uk/pdbe/entry/emdb/EMD-23963 | Electron Microscopy Data Bank, EMD-23963 |
| Khanna K, Lopez-Garrido J, Sugie J, Pogliano K, Villa E | 2021 | Tomogram of a dividing vegetative cell of *Bacillus subtilis* | https://www.ebi.ac.uk/pdbe/entry/emdb/EMD-23964 | Electron Microscopy Data Bank, EMD-23964 |

| | | | | |
|---|---|---|---|---|
| Khanna K, Lopez-Garrido J, Sugie J, Pogliano K, Villa E | 2021 | Tomogram of a dividing vegetative cell of *Bacillus subtilis* FtsZ-linker(Q-rich) strain | https://www.ebi.ac.uk/pdbe/entry/emdb/EMD-23965 | Electron Microscopy Data Bank, EMD-23965 |
| Khanna K, Lopez-Garrido J, Sugie J, Pogliano K, Villa E | 2021 | Tomogram of a dividing sporulating cell of *Bacillus subtilis* | https://www.ebi.ac.uk/pdbe/entry/emdb/EMD-23966 | Electron Microscopy Data Bank, EMD-23966 |
| Khanna K, Lopez-Garrido J, Sugie J, Pogliano K, Villa E | 2021 | Tomogram of a dividing sporulating cell of *Bacillus subtilis* | https://www.ebi.ac.uk/pdbe/entry/emdb/EMD-23967 | Electron Microscopy Data Bank, EMD-23967 |
| Khanna K, Lopez-Garrido J, Sugie J, Pogliano K, Villa E | 2021 | Tomogram of a dividing sporulating cell of *Bacillus subtilis* SpoIIE null mutant | https://www.ebi.ac.uk/pdbe/entry/emdb/EMD-23968 | Electron Microscopy Data Bank, EMD-23968 |
| Khanna K, Lopez-Garrido J, Sugie J, Pogliano K, Villa E | 2021 | Tilt series of dividing vegetative and sporulating cells of *Bacillus subtilis* | https://www.ebi.ac.uk/pdbe/emdb/empiar/EMPIAR-10710 | Electron Microscopy Public Image Archive, EMPIAR-10710 |

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
