## [Decision Letter]

**Acceptance summary:**

This paper from Villa, Pogliano, and colleagues primarily uses cryo-electron tomography to image the division septa of vegetative and sporulating *B. subtilis* cells, finding that the former involve larger, symmetric FtsAZ filaments whereas the latter involve asymmetric filaments. The sporulating septa feature FtsAZ filaments forming on the mother cell side, but not on the forespore side. Prior work had implicated SpoIIE in binding and regulating FtsZ, which this manuscript presents data to further support. The images presented are beautiful and the data convincingly support the cautious claims made, though it remains unclear how the SpoIIE-dependent regulation of FtsAZ filaments in the spore impacts sporulation. The paper also provides important evidence for the field that FtsZ may form continuous rings rather than the discontinuous rings suggested by fluorescence microscopy studies.

**Decision letter after peer review:**

Thank you for submitting your article "Asymmetric localization of the cell division machinery during *Bacillus subtilis* sporulation" for consideration by *eLife*. Your article has been reviewed by 3 peer reviewers, one of whom is a member of our Board of Reviewing Editors, and the evaluation has been overseen by Gisela Storz as the Senior Editor. The following individual involved in review of your submission has agreed to reveal their identity: Ethan C Garner (Reviewer #2).

The reviewers have discussed the reviews with one another and the Reviewing Editor has drafted this decision to help you prepare a revised submission.

As the editors have judged that your manuscript is of interest, but as described below that additional work is required before it is published, we would like to draw your attention to changes in our revision policy that we have made in response to COVID-19 (https://elifesciences.org/articles/57162). First, because many researchers have temporarily lost access to the labs, we will give authors as much time as they need to submit revised manuscripts. We are also offering, if you choose, to post the manuscript to bioRxiv (if it is not already there) along with this decision letter and a formal designation that the manuscript is "in revision at *eLife*". Please let us know if you would like to pursue this option. (If your work is more suitable for medRxiv, you will need to post the preprint yourself, as the mechanisms for us to do so are still in development.)

The reviewers all agreed that this paper presents some fascinating tomograms of *B. subtilis* cells that reveal (i) a potential asymmetry in the FtsZ/A rings formed during sporulation compared to the symmetric rings formed during vegetative growth and (ii) continuity of the FtsZ rings, in contrast to much of the light microscopy-based studies that have concluded Z rings are discontinuous. The work was deemed to be of significant interest to the broad bacterial cell biology field. However, a number of concerns and questions were raised that the reviewers felt should be addressed in a revision.

Although the *eLife* process typically involves the condensing of multiple reviews into a single list of requested changes, in this case, as you'll see, the reviewers each had quite different questions and concerns. So we collectively agreed that it would be best this time to provide the three reviews in their entirety. Most of the issues raised can be dealt with through changes in the text or presentation of data in figures. There are a couple of points raised by Reviewer 3 that suggest experiments – if these have been done and could be included, please do so, but they could alternatively be dealt with by adding appropriate discussion to the text. Reviewer 3 has also raised important concerns about the tomograms presented – to address these concerns, additional tomograms will need to be incorporated and presented in the paper.

*Reviewer #1:*

This paper from Villa, Pogliano, and colleagues primarily uses cryo-electron tomography to image the division septa of vegetative and sporulating *B. subtilis* cells, finding that the former involve larger, symmetric FtsAZ filaments whereas the latter involve asymmetric filaments. The sporulating septa feature FtsAZ filaments forming on the mother cell side, but not on the forespore side. Prior work had implicated SpoIIE in binding and regulating FtsZ, which this manuscript presents data to further support. The images presented are beautiful and the data generally support the cautious claims made. The paper is also very well written and easy to follow. I am, however, left wondering how the SpoIIE-dependent regulation of FtsAZ filaments in the spore impacts sporulation – in other words, the physiological significance of the findings reported were lacking and I wondered whether this could be better addressed.

Why does FtsA form a visible filament in *B. subtilis* but not in *E. coli* or C. crescentus (unless, apparently, FtsA is overexpressed)? The authors briefly mention that the stoichiometry is different, but I guess I'm asking more about what the functional consequences are for such different architectures of FtsA.

If the idea is that only half as much FtsAZ is present at sporulating septa given that it's only present on one side and not the other, why is the septal thickness decreased by 4X? Does this imply something else is contributing to/affecting septal thickness? Lines 185-195 may be explaining this, suggesting that there's also some remodeling of the vegetative septum after closure completes, but I think the authors could clarify things a bit better. And maybe a related point: do the authors think that the post-closure modification is a consequence of the thicker septum or just driven by some remodeling factor only produced during vegetative growth?

Related to the point above, the authors note later, on lines 236-240 that even in the spoIIE mutant the thickness is reduced relative to vegetative septa – is this now just a difference in post-closure remodeling?

Lines 222-23: Here and in the Intro the authors mention 'certain spoIIE mutants' that cause thicker septa. Can you be more specific? What's the nature of these mutants, i.e. internal deletions, point mutants, etc?

It seemed odd not to also examine a variant of SpoIIE lacking only region III to bolster the conclusion that "both the TM and the FtsZ-binding domain are essential to regulate the localization of FtsAZ filaments…" Also, related to the prior point, are there other known spoIIE mutants that localize properly by yield thicker septa that could be examined? And what about overexpressing regions I and II or just region II? Is this sufficient to make septa even thinner? What if such a construct were expressed in mother cells?

Are there mutants of FtsZ that don't interact with SpoIIE?

What's still missing for me at the end of the manuscript is an understanding of why this matters to spores? In other words, what's the physiological importance of blocking FtsAZ assembly in the forespore? Does the spore not mature properly? There are several models noted in the Discussion which I felt should've been experimentally probed to give this manuscript greater impact in terms of the physiological relevance and insight.

The finding noted in the Discussion that Z-rings are continuous in the tomograms, but not in super-resolution microscopy seems important to me and I was surprised this wasn't more fully fleshed out in the Results section. I think it should be, with some detailed description of whether the continuity is entirely 1D or whether lateral interactions/bundling interactions are needed for the continuity observed. And is this seen for vegetative and sporulation septa or do they differ in this regard?

*Reviewer #2:*

The paper Khanna et al. is a novel, insightful, and exciting work, giving several important observations that help unravel the spatial organization and underlying mechanism of sporulation-specific septation. I think the paper is of sufficient quality to be published in *eLife*. However, there are a few small issues that should be addressed first

1. Regarding the FtsA SIM imaging and the subsequent conclusion – "These data suggest that FtsA is present but likely not incorporated into the cell division machinery on the forespore side of the septum".

Given that the septation septum is only 50nm thick and the resolution of SIM is (at best) 120nm, it's hard to believe the FtsA ring they observe in the forespore is associated with the septum at all, as FtsA associated with each side of the septum should not be resolvable from each other with this technique. This comment may have arisen a misinterpretation of the above statement, but currently the "present but not associated" leaves some ambiguity as to whether the FtsA is septum associated, or just present in the forespore. Perhaps more explicit phrasing could correct any confusion.

2. I was a bit surprised the authors consider the FtsZ filament arrangement in the spoIIE-ΔregII strain (Figure 3G-i) to be the same as in the spoIIE deletion strain, as there are some obvious (and perhaps meaningful) differences. In the spoIIE-ΔregII strain, the FtsZ filaments still appear to be off-shifted, but less as dramatically as in wild type. More dramatically, the filaments (per their color annotation) appear to be reduced by about half. Unless other images of this mutant are indeed more similar to wild-type cells, it might be worth noting the differences in this mutant, as to the reader, the spoIIE-ΔregII images kind of jump out as different compared to the other panels. Furthermore, if the reduction in filaments is reproducible, noting this might assist further studies into the role of the FtsZ binding domain of spoIIE in cell division.

3. Likewise, regarding the statement – "Third, our data suggest that during septal biogenesis, the number of FtsAZ filaments tracking the sporulation septum is approximately half of that in the vegetative septum, which likely gives rise to a thinner sporulation septum by mediating the circumferential motion of fewer PG synthases around the division plane during sporulation (Figure 2G, H)" – it should be noted that the spoIIE-ΔregII strain appears to have a reduced number of filaments (per their annotation), yet a similar septal thickness as the spoIIE null. Thus, explaining this discrepancy would help bolster their proposed link between septal thickness and number of FtsZ filaments.

4. The non-FtsA associated filaments along the length of the septa shown in Figure 2—figure supplement 1 panel are very intriguing, but the explanation of how they could remain membrane-associated is a bit vague, and should be further elucidated. Given FtsZ itself cannot associate with the membrane, are the authors proposing the extended linker allow FtsZ filaments to associate with the membrane by associating with each other at a distance, or do the authors think these stray filaments are binding to the membrane through another protein like EzrA or SepF?

*Reviewer #3:*

This study uses electron cryo-tomography (cryo-ET) to image and investigate filament systems in cell division during both vegetative growth and sporulation in *Bacillus subtilis*. This is made possible by the inclusion of cryo FIB milling in the workflow to thin Bacillus cells that would otherwise be too thick for cryo-ET. The authors start by showing that FtsZ and FtsA form membrane distal- and proximal filaments and also that the filaments form small sheets and most likely form a continuous division ring, as previously reported by cryo-ET for *E. coli* and C. crescentus, but in disagreement with super-resolution light microscopy data. The authors then compare the distribution of filaments between vegetative and sporulation septa, and demonstrate that in sporulation septa, the filaments are only found on the mother cell side of the septum and this asymmetry is potentially mediated by SpoIIE. Sporulation septa are significantly thinner than vegetative septa and they propose that this thickness difference is caused by fewer FtsAZ filaments resulting in less peptidoglycan inclusion, possibly mediated by SpoIIE competing with FtsA for FtsZ binding sites. Generally, the work is well presented and clearly a lot of thought has gone into the presentation of figures to make the data understandable to readers who are less familiar with looking at tomography data.

The one-sided asymmetric nature of the filaments in the sporulation septum as depicted in Figure 2C-F is unfortunately not fully convincing as evidenced by the figures presented. The quality of this tomogram is not as good as some of the others and this reviewer finds that the one-sided nature of the filaments is not fully supported by the data as presented. Can more examples of this be shown/be provided? How many times was this observed and was it the case for all tomograms of sporulation septa? The reviewer is aware of the technically demanding nature of the work but the data need to be convincing especially when low-contrast cryo-ET data is being interpreted. As the conclusions from the SpoIIE mutant experiments later in the paper plus the overall conclusions rely heavily on the filaments only being present on the mother cell side I think it is important that the evidence for this is fully convincing.

When tomogram slices are displayed side by side with and without filaments annotated, the annotation of the filaments is sometimes a bit misleading. In the xy views the dots are generally shown as regularly spaced when often this is not the case when examining the densities in the unannotated view, particularly for the membrane proximal filaments. They are also often shown spanning a greater distance than they are convincingly present in the tomogram slice. When viewed in the xz plane the filaments are often depicted as continuous when they are not convincingly, plus extending further in distance than they are present/imaged given the missing wedge issues. In the text it mentions that the filaments are not necessarily continuous, particularly for FtsA and it would be better if the annotation in the figures reflected that in the real data since they are not always perfectly arranged and continuous, nor would this necessarily expected to be the case (as the authors state correctly).

Overall, the work is interesting and provides more evidence for important disagreements in the bacterial cell division field. If more or more convincing evidence can be provided for some of the major conclusions, this work would make a really nice contribution and should be published.

---

## [Author Response]

Reviewer #1:This paper from Villa, Pogliano, and colleagues primarily uses cryo-electron tomography to image the division septa of vegetative and sporulating *B. subtilis* cells, finding that the former involve larger, symmetric FtsAZ filaments whereas the latter involve asymmetric filaments. The sporulating septa feature FtsAZ filaments forming on the mother cell side, but not on the forespore side. Prior work had implicated SpoIIE in binding and regulating FtsZ, which this manuscript presents data to further support. The images presented are beautiful and the data generally support the cautious claims made. The paper is also very well written and easy to follow. I am, however, left wondering how the SpoIIE-dependent regulation of FtsAZ filaments in the spore impacts sporulation – in other words, the physiological significance of the findings reported were lacking and I wondered whether this could be better addressed.

We thank the reviewer for the insightful comments and questions, which have improved our manuscript. We hope that our answers below address their concerns, and that our revised text clarifies these points.

Why does FtsA form a visible filament in *B. subtilis* but not in *E. coli* or C. crescentus (unless, apparently, FtsA is overexpressed)? The authors briefly mention that the stoichiometry is different, but I guess I'm asking more about what the functional consequences are for such different architectures of FtsA.

We thank the reviewer for raising this point. The functional consequences of different architecture of FtsA are not known, but we speculate that it might be related to the different thickness of the septal PG in Gram-positive and Gram-negative bacteria. We have included a paragraph in the discussion addressing this point (lines 361-366): “One possible explanation for this difference between species is that cell division in *B. subtilis* involves the constriction of a septal disc composed of thicker septal PG (~25 nm during sporulation and ~50 nm during vegetative growth) than in *E. coli* or *C. crescentus*, where the septal disc consists of only ~4 nm thick PG. Hence, cytokinesis in *B. subtilis* may require more FtsA filaments in order to tether the increased number of PG synthetases that are required to build a thicker septum during constriction.”

If the idea is that only half as much FtsAZ is present at sporulating septa given that it's only present on one side and not the other, why is the septal thickness decreased by 4X? Does this imply something else is contributing to/affecting septal thickness? Lines 185-195 may be explaining this, suggesting that there's also some remodeling of the vegetative septum after closure completes, but I think the authors could clarify things a bit better. And maybe a related point: do the authors think that the post-closure modification is a consequence of the thicker septum or just driven by some remodeling factor only produced during vegetative growth?

We thank the reviewer for encouraging us to clarify this point. We have expanded the discussion in lines 212-228 to better address it. One likely explanation for the increase in the thickness of the vegetative septum after it is formed is that, during vegetative growth, the septum splits to generate two equal daughter cells upon closure. This process requires the activity of several cell wall hydrolases that cleave PG crosslinks leading to the expansion of the cell wall by distributing forces across the cell surface, which would result in an increase in septal thickness (Huang et al., 2008; Lee and Huang, 2013). Hence, we speculate that during vegetative growth, the increase in the thickness of the septum upon closure is a consequence of septal remodeling due to activity of PG hydrolases. However, during sporulation, the two cells formed (forespore and mother cell) do not split. Rather, one cell (mother cell) engulfs the other (forespore). Hence, remodeling of septal PG by cell wall hydrolases that participate in splitting of the two daughter cells during vegetative growth likely does not happen during sporulation. As a result, the thickness of the sporulation septum after it is formed remains almost the same as during constriction.

Related to the point above, the authors note later, on lines 236-240 that even in the spoIIE mutant the thickness is reduced relative to vegetative septa – is this now just a difference in post-closure remodeling?

The septal thickness of ~42 nm in SpoIIE mutant in the modified lines 275-278 (previously 236-240) corresponds to that of the invaginating septa (not post-closure). We apologize for the confusion and we have included the words “invaginating” wherever appropriate in the text for clarification.

Lines 222-23: Here and in the Intro the authors mention 'certain spoIIE mutants' that cause thicker septa. Can you be more specific? What's the nature of these mutants, i.e. internal deletions, point mutants, etc?

Thanks for the suggestion. We have added a short description of these mutations in lines 257-259: “These mutations correspond to either transposon insertions or point mutations in *spoIIE* locus that do not produce any active gene product and hence represent the null phenotype.”

It seemed odd not to also examine a variant of SpoIIE lacking only region III to bolster the conclusion that "both the TM and the FtsZ-binding domain are essential to regulate the localization of FtsAZ filaments…" Also, related to the prior point, are there other known spoIIE mutants that localize properly by yield thicker septa that could be examined? And what about overexpressing regions I and II or just region II? Is this sufficient to make septa even thinner? What if such a construct were expressed in mother cells?

We agree that examining other SpoIIE mutant strains that lack region III or strains that overexpress regions I and II would be informative. However, due to the demanding resources and time required to perform each of the cryo-ET experiments and the current circumstances, we respectfully find them beyond the scope of this manuscript. However, we have expanded the discussion to better explain the phenotypes associated with mutations in the different SpoIIE regions, specifically region III in lines 474-496 of the Discussion section to better support our conclusion.

Are there mutants of FtsZ that don't interact with SpoIIE?

Currently, we are unaware of any mutants of FtsZ previously reported in the literature that do not interact with SpoIIE. However, strains with mutations in the C-terminus of FtsZ have been previously reported to exhibit several abnormal morphologies and altered interactions with other FtsZ-binding proteins (Feucht and Errington, 2005; Król et al., 2012; Shen and Lutkenhaus, 2009). Testing these strains for their interaction with SpoIIE should further the biochemical basis of interaction of FtsZ and SpoIIE in the future.

What's still missing for me at the end of the manuscript is an understanding of why this matters to spores? In other words, what's the physiological importance of blocking FtsAZ assembly in the forespore? Does the spore not mature properly? There are several models noted in the Discussion which I felt should've been experimentally probed to give this manuscript greater impact in terms of the physiological relevance and insight.

As mentioned by the reviewer, we have speculated on some of the downstream consequences of restricting FtsAZ assembly to the mother cell and the role of a thinner sporulation septa for the completion of engulfment in the Discussion section. Exhaustively testing all possible models in the manuscript is beyond the scope of the manuscript and complicated by several factors. Investigating the physiological relevance of septal thickness during sporulation is confounded by the pleiotropic nature of SpoIIE mutations. SpoIIE directly participates in polar septation, but it is also required for the activation of σ^F^ in the forespore, and consequently, the rest of the sporulation-specific σ factors. Although the two roles seem to depend on different parts of the protein, separating them is not straightforward, as mutations that affect polar septation may also affect the localization of SpoIIE and therefore, the protein might not be released to the forespore to activate σ^F^. In addition, after the polar septum has been formed, several sporulation-specific proteins are produced, some of which form channels across both the septal membranes and some that are involved in PG remodeling (like SpoIIDMP) (as discussed in lines 410-420). These proteins may play additional roles in regulating the septal thickness and physiology. Hence, separating the role of sporulation-specific proteins produced after polar septation from that of IIE in regulating septal thickness is not straightforward, and will likely require thorough genetic analysis. In addition, cryo-FIB-ET experiments are time intensive, challenging and costly, so it is simply not feasible to perform a more comprehensive analysis in this paper.

We made another observation regarding the fate of FtsAZ in the forespore. Time lapse microscopy of FtsA-mNeonGreen and FtsZ-mNeonGreen indicated that just after polar septation, FtsAZ signal in the forespore gradually diminished as engulfment proceeded in these cells (Figure 9—figure supplement 1). It is possible that there are protein(s) in the forespore that mediate degradation of any FtsAZ polymers that don’t participate in cell division but remain behind in the forespore, ensuring an extra layer of safety mechanism to prevent any aberrant cell division events in the forespore. Another recent study from the lab showed that certain proteins involved in central carbon metabolism and metabolic precursor synthesis were depleted specifically in the forespore after polar septation (Riley et al., 2021), although we do not know at this point if the mechanism of depletion of these proteins is similar to that of FtsAZ. Future studies will be required to address this point. We have expanded the Discussion section (lines 438-447) to include these additional insights.

The finding noted in the Discussion that Z-rings are continuous in the tomograms, but not in super-resolution microscopy seems important to me and I was surprised this wasn't more fully fleshed out in the Results section. I think it should be, with some detailed description of whether the continuity is entirely 1D or whether lateral interactions/bundling interactions are needed for the continuity observed. And is this seen for vegetative and sporulation septa or do they differ in this regard?

We thank the reviewer for the comment. We have now modified the Results section in lines 165-176 to discuss the continuity of Z-rings in vegetative cells. We have also shown the distribution of intensity values for the region traced by the Z-and A-rings in Figure 3—figure supplement 2 that suggest the more continuous nature of Z-ring compared to the A-ring during vegetative growth. During sporulation, we notice that the septum is not as sharp as during vegetative growth, probably due to molecular crowding by membrane proteins that participate in cell division as well as those that are produced specifically during sporulation (Figure 7). Also, since FtsAZ are only present on the mother cell side and the distance spanned by the filament bundle is much shorter, the filaments are detected in only a few slices in cross-sectional views as opposed to vegetative growth. These limitations make it difficult to conclude with confidence whether the Z-ring is continuous or not during sporulation (included in lines 184-189 in the Results section). However, the Z-rings appear continuous during sporulation in SpoIIE sporangia in the tomograms we acquired (Figure 8), suggesting their continuous nature in at least under this condition during sporulation. We have expanded the Discussion section (lines 391400) to include these insights.

Reviewer #2:The paper Khanna et al. is a novel, insightful, and exciting work, giving several important observations that help unravel the spatial organization and underlying mechanism of sporulation-specific septation. I think the paper is of sufficient quality to be published in eLife. However, there are a few small issues that should be addressed first1. Regarding the FtsA SIM imaging and the subsequent conclusion – "These data suggest that FtsA is present but likely not incorporated into the cell division machinery on the forespore side of the septum".Given that the septation septum is only 50nm thick and the resolution of SIM is (at best) 120nm, it's hard to believe the FtsA ring they observe in the forespore is associated with the septum at all, as FtsA associated with each side of the septum should not be resolvable from each other with this technique. This comment may have arisen a misinterpretation of the above statement, but currently the "present but not associated" leaves some ambiguity as to whether the FtsA is septum associated, or just present in the forespore. Perhaps more explicit phrasing could correct any confusion.

We have modified the description in the lines 309 – 325 to avoid any confusion regarding the interpretation of SIM and time-lapse imaging data.

2. I was a bit surprised the authors consider the FtsZ filament arrangement in the spoIIE-ΔregII strain (Figure 3G-i) to be the same as in the spoIIE deletion strain, as there are some obvious (and perhaps meaningful) differences. In the spoIIE-ΔregII strain, the FtsZ filaments still appear to be off-shifted, but less as dramatically as in wild type. More dramatically, the filaments (per their color annotation) appear to be reduced by about half. Unless other images of this mutant are indeed more similar to wild-type cells, it might be worth noting the differences in this mutant, as to the reader, the spoIIE-ΔregII images kind of jump out as different compared to the other panels. Furthermore, if the reduction in filaments is reproducible, noting this might assist further studies into the role of the FtsZ binding domain of spoIIE in cell division.

For the previous Figure 3G-I (now modified Figure 8D-F), we showed a slice from the middle of the tomogram. This particular tomogram was from a very thin lamella (~70 nm thickness) and hence some of the densities corresponding to FtsZ filaments were not visible in the representative slice in the figure panel. We have now added a supplemental figure, Figure 8—figure supplement 3 wherein some of the FtsZ densities are also visible on the mother cell side in another slice for the same tomogram shown in the now modified Figure 8D-F. We have also added a movie showing the different slices of the tomogram (Video 4). In addition, we have added another tomogram in Figure 8—figure supplement 4 for the *spoIIE-ΔregII* strain.

3. Likewise, regarding the statement – "Third, our data suggest that during septal biogenesis, the number of FtsAZ filaments tracking the sporulation septum is approximately half of that in the vegetative septum, which likely gives rise to a thinner sporulation septum by mediating the circumferential motion of fewer PG synthases around the division plane during sporulation (Figure 2G, H)" – it should be noted that the spoIIE-ΔregII strain appears to have a reduced number of filaments (per their annotation), yet a similar septal thickness as the spoIIE null. Thus, explaining this discrepancy would help bolster their proposed link between septal thickness and number of FtsZ filaments.

We have addressed this concern in comment 2 above.

4. The non-FtsA associated filaments along the length of the septa shown in Figure 2—figure supplement 1 panel are very intriguing, but the explanation of how they could remain membrane-associated is a bit vague, and should be further elucidated. Given FtsZ itself cannot associate with the membrane, are the authors proposing the extended linker allow FtsZ filaments to associate with the membrane by associating with each other at a distance, or do the authors think these stray filaments are binding to the membrane through another protein like EzrA or SepF?

Thank you for the suggestion. We have elaborated the description for these stray filaments and how they may anchor to the membrane in the lines 142-147 of the revised manuscript as follows: “This is substantiated by previous studies showing that *B. subtilis* can grow without FtsA, although albeit slowly and in a filamentous manner (Beall and Lutkenhaus, 1992). Instead, FtsZ may use an alternate anchor like SepF or EzrA to bind to the membrane in the absence of FtsA as these divisome proteins also have an N-terminal domain that binds to the membrane and a C-terminal domain that interacts with FtsZ (Duman et al., 2013; Singh et al., 2007).”

Reviewer #3:[…] The one-sided asymmetric nature of the filaments in the sporulation septum as depicted in Figure 2C-F is unfortunately not fully convincing as evidenced by the figures presented. The quality of this tomogram is not as good as some of the others and this reviewer finds that the one-sided nature of the filaments is not fully supported by the data as presented. Can more examples of this be shown/be provided? How many times was this observed and was it the case for all tomograms of sporulation septa? The reviewer is aware of the technically demanding nature of the work but the data need to be convincing especially when low-contrast cryo-ET data is being interpreted. As the conclusions from the SpoIIE mutant experiments later in the paper plus the overall conclusions rely heavily on the filaments only being present on the mother cell side I think it is important that the evidence for this is fully convincing.

We thank the reviewer for the comment. For the particular tomogram shown in the previous Figure 2C-F (now modified Figure 7A-C), the tomogram by itself is of good quality compared to others we have shown in the manuscript, but we found that the membranes at the leading edge of the nascent septum in sporulating cells are not as defined as in vegetative cells or in SpoIIE mutants. We have discussed this in the lines 184-189 in the Results section. In summary, in addition to the presence of transmembrane cell division proteins, several sporulation-specific multipass transmembrane proteins like SpoIIE, SpoIIGA, and SpoIIIE are present at the invaginating septal membrane during sporulation that increase the ratio of protein to membrane at the septal membrane. This makes the bilayer less sharp compared to its in other conditions (like vegetative growth) that we imaged.

We acquired a total of 11 tomograms of dividing sporulating cells. Of these, we detected the filaments and their localization in mother cell in both the top view and the rotated view clearly in 4 tomograms. We have shown two of them in Figure 7—figure supplement 2,3. Of the remaining tomograms, 1 was too thick and 2 had just started constricting, and we couldn’t detect any clear FtsAZ filaments/dots in these 3. The remaining 4 tomograms were on the verge of completing septum formation. At these later stages of polar septum formation, we clearly detected FtsAZ filaments in top views in 2 of the tomograms (one instance shown in Figure 7—figure supplement 4). Likely, the filamentous rings were closing in at this point and some FtsAZ subunits may have already started to localize to the opposite pole to form the second septum or may have been disassembled. The other 2 of the 4 tomograms were at a very late stage of septum constriction with a gap of ~60-70 nm between the left and the right septa in the middle slice and we could not detect any filaments/dots corresponding to FtsAZ at this stage. We also observed FtsAZ dots/filaments localized only in the mother cell side of the dividing septa in two mutant sporangia – SpoIIB and SpoIIIE^ATP-^ (tomogram corresponding to SpoIIIE^ATP-^ shown in Figure 7—figure supplement 5). Although both SpoIIB and SpoIIIE localize to the leading edges of the invaginating septa during sporulation, there is no evidence of direction interaction between FtsZ and either of these two proteins, suggesting that the localization of FtsAZ filaments in SpoIIB and SpoIII^ATP-^ mutant sporangia should mimic that of wild type. Hence, every time we detected FtsAZ filaments in SpoIIE+ sporulating septa, they were associated with the mother-cell side of the septum. Taking all the above data into account, we believe that our observation that during sporulation, FtsAZ filaments are present on the mother cell side of the invaginating septum stands firm.

When tomogram slices are displayed side by side with and without filaments annotated, the annotation of the filaments is sometimes a bit misleading. In the xy views the dots are generally shown as regularly spaced when often this is not the case when examining the densities in the unannotated view, particularly for the membrane proximal filaments. They are also often shown spanning a greater distance than they are convincingly present in the tomogram slice. When viewed in the xz plane the filaments are often depicted as continuous when they are not convincingly, plus extending further in distance than they are present/imaged given the missing wedge issues. In the text it mentions that the filaments are not necessarily continuous, particularly for FtsA and it would be better if the annotation in the figures reflected that in the real data since they are not always perfectly arranged and continuous, nor would this necessarily expected to be the case (as the authors state correctly).

Our rationale behind showing two slices, one annotated and another unannotated, was to guide readers who are not familiar with studying cryo-ET data, while letting them compare the two slices to look at the cytoskeletal filaments in a given location of the tomogram. In the process, we inadvertently annotated slices/locations that did not point directly to each filament in the xy slices (as dots) or xz/yz rotated views (as filaments – continuous/non-continuous). We have now made sure to highlight FtsZ dots (membrane-distal) correctly in all the xy views. Further, where we could not resolve distinctive dots or filaments, we have depicted them as dashed lines. As a consequence, to ensure that data is not overinterpreted, FtsA filaments (membrane-proximal) in xy slices are often showed as dashed lines instead of dots where it wasn’t possible to resolve individual cross sections of filaments (dots) with confidence. To represent the cytoskeletal filaments in rotated views (xz/yz slices), we now use dashed lines (instead of continuous lines) for both FtsA and FtsZ filaments as assigning precise continuity at the pixel levels for all images is limited by the nature of the data (i.e., missing wedge and signal to noise ratio). We have also taken care to not overextend the annotation of the filaments. We thank the reviewer for this valuable comment. We have made a note for these annotations in Figure legend 3.